# Comparison of Antarctic polar stratospheric cloud observations by ground-based and spaceborne lidars and relevance for Chemistry Climate Models

Marcel Snels[1], Andrea Scoccione[1], Luca Di Liberto[1], Francesco Colao[2], Michael Pitts[3], Lamont Poole[4], Terry Deshler[5], Francesco Cairo[1], Chiara Cagnazzo[1], and Federico Fierli[1]

[1]Istituto di Scienze dell'Atmosfera e del Clima, Via Fosso del Cavaliere 100, 00133 Roma
[2]ENEA, Via Enrico Fermi 45, 00044 Frascati
[3]NASA Langley Research Center, Hampton, Virginia 23681, USA
[4]Science Systems and Applications, Inc., Hampton, Virginia, 23666, USA
[5]University of Wyoming, Laramie, WY 82071, USA

*Correspondence to:* Marcel Snels (m.snels@isac.cnr.it)

**Abstract.** A comparison of polar stratospheric clouds (PSCs) occurrence from 2006 to 2010 is presented, as observed from the ground-based lidar station at McMurdo (Antarctica) and by the satellite-borne CALIOP lidar (Cloud-Aerosol Lidar with Orthogonal Polarization) measuring over McMurdo. McMurdo (Antarctica) is a primary lidar station for aerosol measurements of the NDACC (Network for Detection of Atmospheric Climate Change). The ground-based observations have been classified
with an algorithm derived from the recent v2 detection and classification scheme, used to classify PSCs observed by CALIOP.

A statistical approach has been used to compare ground-based and satellite based observations, since point-to-point comparison is often troublesome due to the intrinsic differences in the observation geometries and the imperfect overlap of the observed areas.

A comparison of space-borne lidar observations and a selection of simulations obtained from Chemistry Climate Models has
been made by using a series of quantitative diagnostics based on the statistical occurrence of different PSC types. The distribution of PSCs over Antarctica, calculated by several CCMVal-2 and CCMI chemistry climate models has been compared with the PSC coverage observed by the satellite borne CALIOP lidar. The use of several diagnostic tools, including the temperature dependence of the PSC occurrences, evidences the merits and flaws of the different models. The diagnostic methods have been defined to overcome (at least partially) the possible differences due to the resolution of the models and to identify differences
due to microphysics (e.g. the dependence of PSC occurrence on T-$T_{NAT}$).

A significant temperature bias of most models has been observed as well as a limited ability to reproduce the longitudinal variations in PSC occurrences observed by CALIOP. In particular a strong temperature bias has been observed in CCMVal-2 models with a strong impact on PSC formation. The WACCM-CCMI (Whole Atmosphere Community Climate Model - Chemistry Climate Model Initiative) model compares rather well with the CALIOP observations, although a temperature bias
is still present.

## 1 Introduction

Lidar observations have been extensively used to characterize the occurrence of PSCs in the polar stratosphere (see e.g.Browell et al. (1990); Adriani et al. (2004); Di Liberto et al. (2014); Achtert and Tesche (2014)) . The observed optical parameters allow
to discriminate different cloud types, such as STS (supercooled ternary solution), NAT (nitric acid trihydrate) and water ice, and external mixtures of the former. Pitts and co-workers (Pitts et al., 2009, 2011, 2013, 2018), calculated the optical parameters of cloud particles with different size distributions and chemical composition in order to define a PSC classification, which was then applied to the CALIOP data. Achtert and Tesche (Achtert and Tesche, 2014) made an assessment of several lidar-based PSC classifications and their impact on the occurrences of the different PSC types. Their conclusion was that the comparison of PSC
classifications obtained from different lidar observations is not straightforward and should take into account the measurement technique and classification methodology used. A variety of schemes using different thresholds for detection and classification have been proposed, rendering a comparison difficult. Here we want to compare ground-based and satellite based lidar data, by using a detection and classification scheme for the ground-based data, which closely approaches the new v2 classification scheme used for CALIOP (Pitts et al., 2018).

Ground-based lidar observatories provide a unique data base, having decadal coverage, albeit with discontinuities, spanning from the middle eighties to today.

The first lidar observations in Antarctica started in 1985 at Syowa Station. Iwasaka and co-workers (Iwasaka, 1985, 1986) used a polarization sensitive lidar to measure backscatter and depolarization to observe PSCs. Later, in 1987/1988, at the Amundsen-Scott South Pole Station, Fiocco and co-workers (Fiocco et al., 1992) used the elastic backscatter signal from a
lidar operating at 532 nm to observe PSCs in relation to the temperature. PSCs have also been observed at Davis, from 2001 to 2004 (Innis and Klekociuk, 2006) and at Rothera (Simpson et al., 2005) from 2002 to 2005.

Long-term observations of PSCs have been performed at McMurdo (Adriani et al., 1992, 1995, 2004; Di Liberto et al., 2014), from 1989 until 2010 and at Dumon D'Urville (Santacesaria et al., 2001; David et al., 1998, 2010), from 1990 until now, both with polarization sensitive lidars. Recently the McMurdo lidar has been transferred to Dome C and is operating there
from 2014 on (Snels et al., 2018).

A clear issue is that the representativeness of ground-based long-term lidar data series of the Antarctic stratosphere might limit their value in climatological studies and model evaluation. Since the long-term ground based lidar observations have been performed only in few locations, the comparison with model simulations and satellite borne instruments is necessarily limited to these locations, which poses a limit to their use. The recent availability of satellite-borne lidar observations provides an almost
complete coverage of the globe, and presents the opportunity to test the polar stratospheric cloud scheme of Chemistry Climate Models (CCMs) on synoptic scales. The Cloud-Aerosol Lidar and Infrared Pathfinder Satellite Observations (CALIPSO) was launched in April 2006 with the primary objective of improving our understanding about the impact of clouds and aerosol on the climate. CALIOP provides total backscatter and depolarization profiles, allowing classification of the observed clouds and

aerosols. The original CALIPSO mission had a minimum time frame of 3 years, but has been extended several times and is still active.

Comparison between CALIOP and ground-based observations in the Antarctic stratosphere of PSCs is thus possible from 2006 on and has been pursued in the case of McMurdo Station by performing co-incident measurements with CALIPSO overpasses whenever possible.

Due to their primary role in ozone chemistry, a correct representation of PSCs in CCMs is needed. Actually, the parametrization of PSC formation in most CCMs depends only on temperature thresholds and on nitric acid and water vapour concentrations for the determination of supersaturation conditions. A rather complete description of the parametrizations used in state-of-the-art CCMs is reported in Morgenstern et al. (2017). The SPARC Report N$^o$5 (2010) Chemistry-Climate Model Validation (CCMVal-2) (Eyring et al., 2010) has shown that Chemistry Climate Models can have a biased representation of the stratospheric conditions with colder temperatures that lead to an overestimate of ozone depletion, also due to an unrealistic PSC coverage. Hence PSC simulations show a large uncertainty, as reported in the CCMVal-2 report. Nevertheless, the report presents a preliminary evaluation based on global averages with a subset of CALIOP data.

The most recent CCMs are able to reproduce the denitrification by the formation of STS and NAT and the dehydration through the formation of ice clouds, but use rather approximate schemes based on temperature thresholds for the onset of nucleation, with additional constraints on how much of the available nitric acid is depleted by STS and NAT formation. Although the overall denitrification and dehydration can be represented rather well, the correct description of the formation of STS and NAT, and mixed type PSCs would need a more sophisticated microphysics model.

In the present work we first compare the statistics of occurrence of different PSC classes in the stratosphere over McMurdo Station, as detected by the ground-based lidar operating there and the satellite-borne CALIOP. Subsequently we use the full coverage of the Antarctic CALIOP data to assess the performances of different CCMs in simulating PSC occurrences and PSC distribution over Antarctica.

## 2 Comparison of PSC observations by ground-based and satellite based lidars

### 2.1 CALIPSO PSC observations

The CALIPSO satellite was launched in April 2006 as a component of the A-train satellite constellation (Stephens et al., 2002, 2017). With an orbit inclination of 98.2 °, it provides extensive daily measurement coverage over the polar regions of both hemispheres, up to 82 ° in latitude. It hosts the CALIOP two wavelength polarization diversity lidar, that measures backscatter at wavelengths of 1064 nm and 532 nm, the latter signal separated into parallel and cross polarization, with respect to the polarization of the outgoing laser beam. Details of CALIOP can be found in Hunt et al. (2009) and Winker et al. (2009). CALIOP data have extensively been used for observing PSCs and improved algorithms for PSC classification have been reported in Pitts et al. (2009, 2011, 2013, 2018).

## 2.2 Ground-based PSC observations at McMurdo

A Rayleigh polarization diversity lidar has operated in the Antarctic station of McMurdo since 1991, in the framework of an USA-Italian collaboration (Adriani et al., 2004; Di Liberto et al., 2014). It measures aerosol backscatter and depolarization profiles from 12 km to 30 km, with a vertical resolution of 30 meters. Aerosol backscattering is retrieved using the Klett algorithm (Klett, 1981) and the extinction is calculated according to Gobbi (1995). The depolarization is calibrated following the method described in Snels et al. (2009). The lidar was operated by science technicians of the National Science Foundation (NSF) during the Antarctic winter, typically from the end of May until the end of September to cover the whole period of PSC occurrence. Potential vorticity reanalysis shows that McMurdo is well within the stratospheric polar vortex from mid-June to the end of September, except for rare events of major vortex perturbation. As a routine, the lidar is operated at the same time every day when meteorological conditions are favorable, or at the earliest chance to do so, for about 30 minutes to render a single profile. When possible, the observations are synchronized with overpasses of the CALIPSO satellite, when its footprint is within 100 km distance from McMurdo. Observations are intensified in coincidence with Optical Particle Counter (OPC) and ozone sondes balloon measurements (Adriani et al., 1992). All observations at a wavelength of 532 nm used in the present analysis have been quality checked and the relevant data are publicly available in the NDACC data base (ftp://ftp.cpc.ncep.noaa.gov/ndacc/station/mcmurdo/ames/lidar/).

For the ground-based lidar data a single vertical profile with a vertical resolution of 150 m is obtained by averaging 30 minutes of acquisition.

## 2.3 PSC detection and classification

PSC detection and classification from lidar measurements with orthogonal polarization is usually based on two optical parameters derived from the optical signals with parallel and perpendicular polarization with respect to the laser, the backscatter ratio and the aerosol depolarization. Here we use the backscatter ratio $R$ and the perpendicular backscatter coefficient $\beta_\perp$, in order to be consistent with the v2 detection and classification scheme used for the CALIOP data. The backscatter ratio is defined as

$$R = (\beta^{aer} + \beta^{mol})/\beta^{mol} \tag{1}$$

where $\beta^{aer}$ is the total aerosol backscatter and $\beta^{mol}$ is the total molecular backscatter.

We must bear in mind that for all lidar measurements the optical parameters represent an average value of the microscopic properties of an ensemble of many particles in a large air volume which may belong to different composition classes. Only rarely the observation of an air volume can be totally attributed to a single class of particles, except for particular cases where the temperature conditions exclude the co-existence of particles with different compositions. Thus the resulting macroscopic optical parameters are mostly due to external mixtures and are dominated by the species with the largest relative abundance and/or the largest optical parameters. When classifying the PSCs, the classifications indicate the dominant species or the mixtures of species.

## 2.4 PSC Detection and classification criteria for the CALIPSO v2 data

The CALIOP v2 PSC detection and composition classification algorithm (Pitts et al., 2018) has been used to create the recently released CALIOP v2 PSC mask database covering the period from June 2006 to October 2017. Here we compare these v2 data with ground-based observations at McMurdo from 2006 to 2010. Major enhancements in the v2 algorithm over earlier versions include daily adjustment of composition boundaries to account for effects of denitrification and dehydration, and estimates of the random uncertainties $u(\beta_{\perp})$ and $u(R)$ due to shot noise in each data sample, which are used to establish dynamic detection thresholds and composition boundaries. The CALIOP v2 algorithm is represented pictorially in Figure 1 and is described in more detail in the following sections.

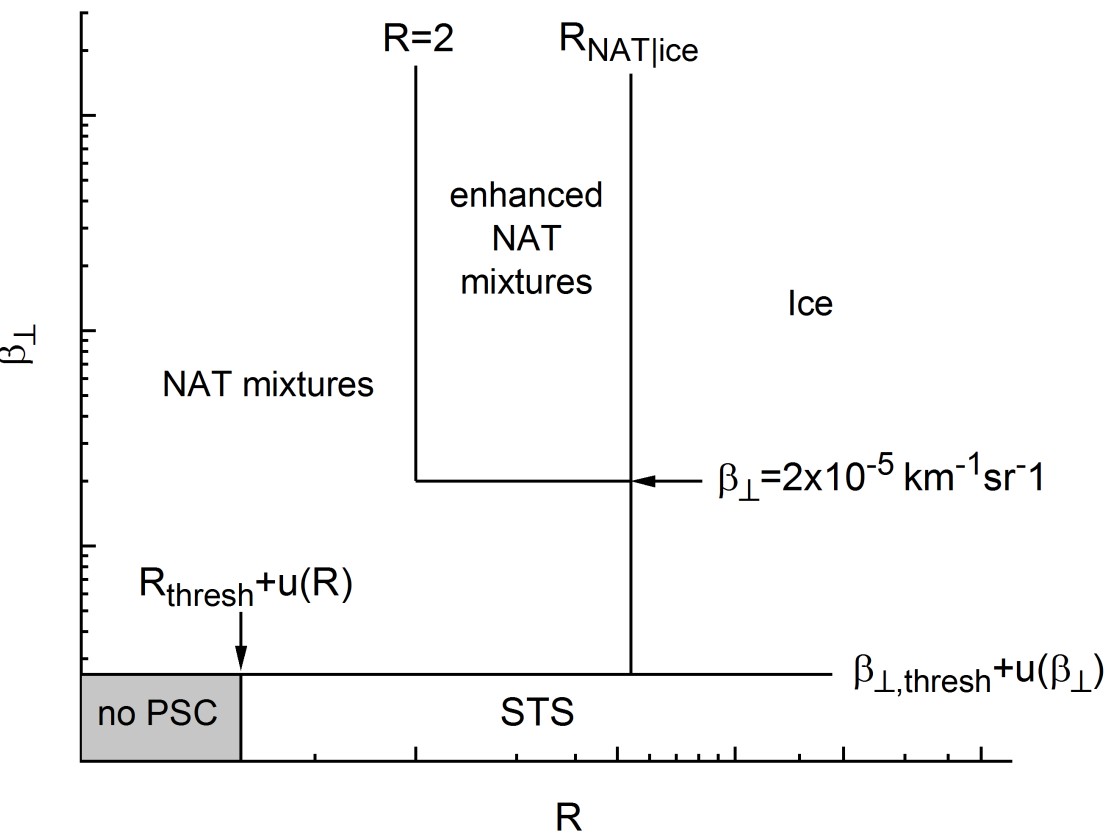

**Figure 1.** The figure shows the detection and classification criteria of the V2 CALIOP algorithm. The classification as STS, NAT mixtures, enhanced NAT mixtures and ice, requires that threshold conditions for $R$ and/or $\beta_{\perp}$ are satisfied. See the text for details.

### 2.4.1 PSC detection

PSCs are detected in the CALIOP data as statistical outliers relative to the background stratospheric aerosol population. The v2 background aerosol thresholds $\beta_{\perp,thresh}$ and $R_{thresh}$ are calculated as the daily median plus one median deviation of CALIOP data at ambient temperatures above 200 K. PSCs are those data points for which either $\beta_{\perp} > \beta_{\perp,thresh}$+u( $\beta_{\perp}$) or $R > R_{thresh}$ +u($R$). If $\beta_{\perp} \leq \beta_{\perp,thresh}$ +u( $\beta_{\perp}$) and $R \leq R_{thresh}$ +u($R$), the point is a non-PSC. Noise spikes are eliminated in the CALIOP v2 data by requiring coherence within a running 3-point vertical by 5-point horizontal along-track box.

### 2.4.2 PSC composition

The PSC composition is determined as follows:

- If $\beta_{\perp} \leq \beta_{\perp,thresh}$ +u( $\beta_{\perp}$), but $R > R_{thresh}$ +u($R$), the PSC is classified as STS.

- A PSC with $\beta_{\perp} > \beta_{\perp,thresh}$+u( $\beta_{\perp}$) is assumed to contain non-spherical particles and is classified as NAT (or enhanced NAT) mixture or ice based on its value of $R$. The boundary value separating ice from NAT and enhanced NAT mixtures, $R_{NAT|ice}$, is calculated based on the total abundances of $HNO_3$ and $H_2O$ vapors as determined on a daily basis as a function of altitude and equivalent latitude from nearly coincident cloud-free Aura MLS data.

- If $\beta_{\perp} > \beta_{\perp,thresh}$+u( $\beta_{\perp}$) and $R > R_{NAT|ice}$, the PSC is classified as ice.

- If $2 < R < R_{NAT|ice}$ and $\beta_{\perp} > 2\cdot10^{-5}$ m$^{-1}$sr$^{-1}$, the PSC is classified an enhanced NAT mixture. All other PSCs with $\beta_{\perp} > \beta_{\perp thresh}$ +u( $\beta_{\perp}$) and $R < R_{NAT|ice}$ are classified as NAT mixtures.

The CALIOP v2 data set provides both the grid of classified PSCs according to the v2 algorithm and the associated optical parameters.

### 2.5 PSC Detection and classification criteria for the ground-based data

In order to compare the ground based lidar data to the CALIOP data we have adopted a new algorithm which follows the same approach and uses the same optical parameters as the v2 CALIOP algorithm (see Figure 1 ).

### 2.5.1 PSC detection

The ground-based raw data have been re-elaborated to produce the backscatter ratio $R$ and the perpendicular backscatter coefficient $\beta_{\perp}$. While the determination of the background aerosol thresholds for the CALIOP data uses a very large number of observations, the quantity of ground-based lidar data is much smaller and does not allow a similar treatment. Instead of using daily medians we calculated a median value from all ground-based data in the 5-year period without PSCs (typically before 15 June or after 1 October) or in obvious clear sky conditions. Thus the background aerosol thresholds were determined as the median values plus one standard deviation about the median. In this way we obtained fixed background thresholds for the backscatter ratio $R_{thres}$ =1.15, and also for $\beta_{\perp}$ =1·10$^{-6}$ m$^{-1}$sr$^{-1}$. While most PSC detection schemes for ground-based lidar data use a threshold only for $R$ (Achtert and Tesche, 2014), the scheme used here is more permissive and allows all data with $R$ >1.15 +u($R$) or $\beta_{\perp,thresh} > 1\cdot10^{-6}$ m$^{-1}$sr$^{-1}$+u( $\beta_{\perp}$), where u($R$) and u($\beta_{\perp}$) are the running standard deviations over altitude,

and a local temperature below 200 K in a range between 12 and 30 km to be detected as PSCs. Note that this procedure is very similar to the v2 CALIOP algorithm, except that we use fixed background thresholds and different estimates of the uncertainties in the data. Finally, to mimic the CALIOP coherence criteria, we require continuity along the vertical profile to avoid identifying isolated noise spikes as PSCs.

### 2.5.2 PSC composition

Composition classification for ground-based PSCs is nearly identical to the CALIOP v2 procedure, the exception being that we use monthly averages for $R_{NAT|ice}$ computed from daily values included in the v2 CALIOP data files.

## 2.6 Comparison of co-located PSC observations at McMurdo from the ground and from CALIPSO during the 5-year observation period

Here we compare PSC statistics from ground-based and satellite-borne lidars, with the goal to assess if the differing measurement procedures used for each of them, induce a bias in the PSC classification, which might hamper the definitions of useful common diagnostics for assessing the performance of regional and global models.

A comparison is made using 248 profiles acquired by the ground-based lidar and 585 overpasses extracted from the CALIOP data base within a $7°$x$2°$ longitude-latitude box centered on the McMurdo site for the years 2006-2010. The choice of the box dimension is dictated by the need to have a minimal latitudinal range (to avoid the inclusion of data in different vortex regimes for stations close to the polar vortex edge) and to have a significant number of observations (a larger longitudinal range assuming a local uniform distribution around the site) and corresponds roughly with a distance of 100 km from McMurdo.

CALIOP overpasses do not occur every day and at most twice per day. In average we have up to 40 CALIOP overpasses per month. Ground-based lidar data are mostly recorded during a CALIOP overpass, but also on days without CALIOP overpasses, usually at the same time that CALIOP overpasses occur and sometimes at different times from the CALIOP overpasses. The latter are not included in this analysis. All other ground-based measurements have been used in the statistical comparison. Generally speaking most of the ground-based profiles have been recorded during a CALIOP overpass, but there might be days with either a ground-based measurement or a CALIOP measurement. So we include all CALIOP measurements falling in a spatial box around McMurdo, and all ground-based data measured in a time frame dictated by CALIOP overpasses, including also the days without overpass.

The comparison between data obtained by space-borne and ground-based instruments is not straightforward. Lidars on satellites provide altitude resolved PSC observations on a synoptic scale, with fixed revisit times on the ground spot, and their observations in the stratosphere are unaffected by tropospheric visibility. Ground-based observations are limited by the weather conditions and become prohibitive in case of heavy cloud cover. Moreover the measurements occur once or twice per day, possibly in co-incidence with satellite overpasses. Sometimes they are conditioned by other activities such as intensive measurement campaigns of other instruments. The different geometry and measurement protocols might induce a bias in PSC statistics of ground-based and satellite-based lidar observations.

The ground-based lidar observes at distances up to 30 km from the ground, while the satellite based lidar is in orbit at 705 km and observes backscattering from distances around 700 km. This implies that the signal-to-noise ratio of CALIOP is in general lower than that of the ground-based lidar. Therefore the CALIOP data use averaging processes where the signal-to-noise ratio is low, and varies the threshold on both $R$ and $\beta_\perp$ as a function of signal-to-noise ratio.

For these reasons, a point-to-point profile comparison of these data bases may not be sufficient to evaluate whether or not the instruments provide a compatible information of PSCs coverage and partition in different classes, which, at the end is the information needed to evaluate models and provide a climatic survey of the polar stratosphere.

The purpose of this analysis is not to perform a validation of the satellite-borne instrument, but to verify if the two instruments provide compatible information in terms of occurrences of the different PSC classes around McMurdo.

In order to illustrate how ground-based and space-borne lidar observations of PSCs compare, we show as an example the height-time evolution of PSC classes for CALIPSO and McMurdo data bases for the year 2006 (see Figure 2), having the best temporal coverage with respect to the other years (2007-2010).

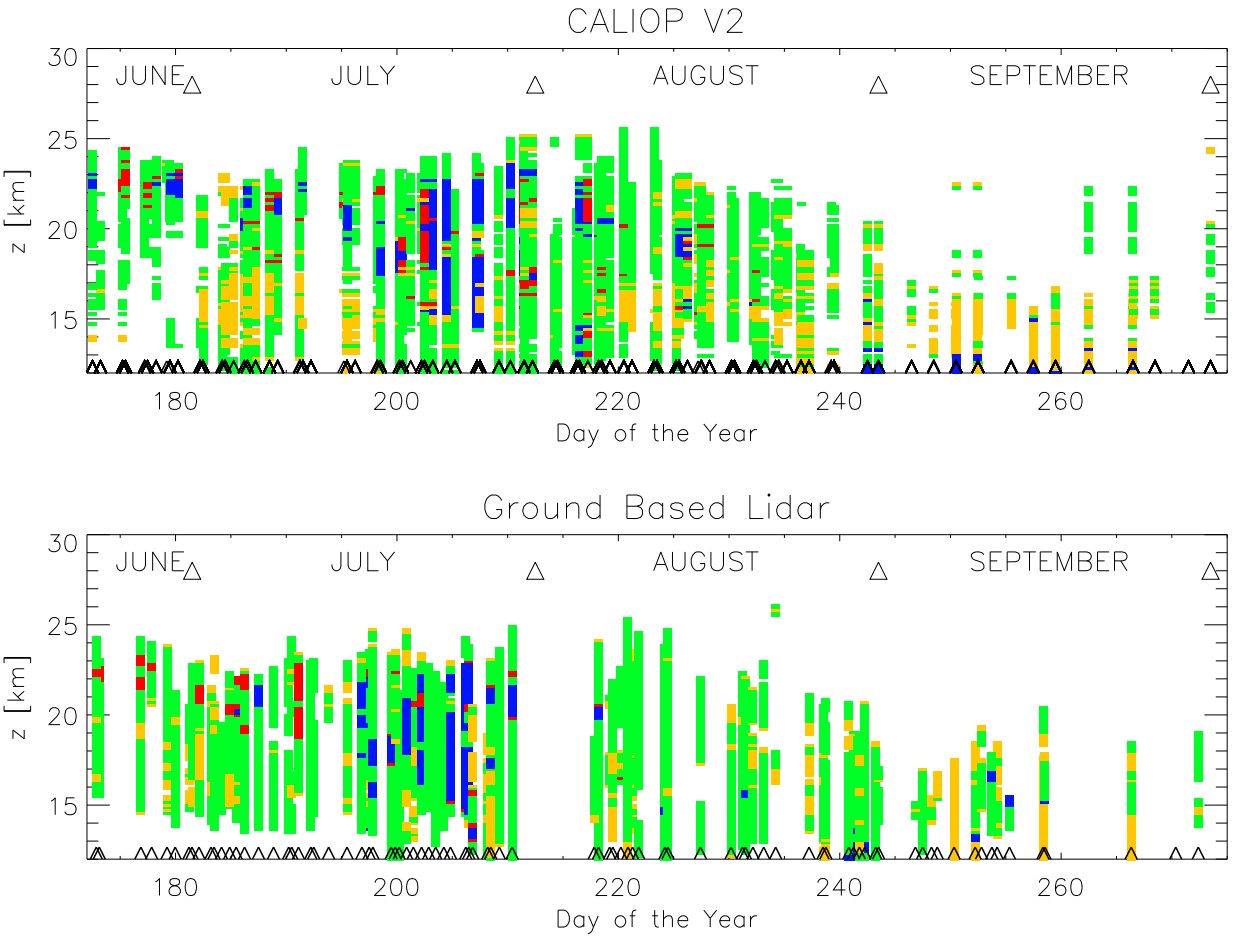

**Figure 2.** PSC observations recorded in 2006 above McMurdo. Upper panel: CALIOP around McMurdo from v2 product. Lower panel: Ground-based lidar data. The PSC classes are represented by colors; green = NAT mixtures, orange = STS, blue = ice, red = enhanced NAT mixtures. Triangles on the x-axis indicate the day when at least one observation was available.

5        Both the CALIOP PSC product, and the classification of the ground-based lidar optical parameters, obtained with the v2 algorithm adapted for ground-based data, provide a similar view for this winter with a dominance of NAT mixtures with isolated periods of ice PSCs in July. Enhanced NAT mixtures appear mostly in June and July, around and above 20 km, while STS has been observed in the lower layers throughout the season, being the major species in September. These results are not directly comparable with the analysis previously reported (Di Liberto et al., 2014), where a different classification scheme for ground-based data was adopted and different PSC classes were assigned. Although the overall agreement with CALIOP is acceptable, many small differences are evident, and confirm that a point-to-point comparison of these data is not straightforward.

For this reason a statistical comparison, including all measurements of a specific Antarctic winter, from 2006 to 2010, has been pursued. This statistical comparison is meaningful as long there is a good coverage in time. Here we show the results

|  | 2006 | | 2006-2010 | |
| --- | --- | --- | --- | --- |
| PSC classes | CALIOP | gr.based | CALIOP | gr.based |
| STS | 15.5 | 14.6 | 22.4 | 13.8 |
| NAT mixtures | 73.6 | 76.0 | 60.1 | 71.6 |
| enhanced NAT mixtures | 2.3 | 2.4 | 2.5 | 2.6 |
| ice | 8.5 | 7.1 | 15.0 | 12.0 |
| overpasses/observations | 128 | 75 | 615 | 248 |

**Table 1.** Frequency of occurrence (in %) of PSC classes for 2006 and for 2006-2010. The last line represents the number of overpasses in the McMurdo box for CALIOP and the number of ground-based observations. Observations in the 12-30 km interval have been considered. The ice-class for CALIOP includes also mountain wave ice.

5    only for 2006, being the year with the best coverage in the ground-based data set. Table 1 shows the number of occurrences for each PSC species for 2006 and the full period of 5 years. A point-to-point comparison might be approximated by a statistical analysis of the data for the shortest possible period. For 2006 the number of profiles for July and August might be sufficient to perform a month by month comparison, although with a larger uncertainty due to the smaller number of observations. In table 2 the occurrences for the PSC classes have been calculated per month for 2006. The agreement is reasonable for STS and NAT

10   mixtures, which account for 80 to 90 % of the observed PScs. The sum of ice and enhanced NAT mixtures shows also a good agreement, while the repartition between the two classes shows some differences. This might be due to the fact that the value of $R_{NAT|ice}$, which separates the two classes might be different for CALIOP and ground-based data set. The extrapolation of $R_{NAT|ice}$ from the CALIOP dataset might be poor, because of the distance of the overpass track with respect to the location of the ground station. The maximum deviation in occurrences between CALIOP and ground-based observations is in the order

15   of 5 %, which is acceptable, considering all possible biases.

We also compare the PSC occurrences as a function of altitude during the winter season, by accumulating all PSC observations each month of 2006 (July and August) between 12 and 30 km. In figure 3 the vertical profiles of monthly PSC occurrence for 2006 are reported. Occurrence is calculated as the fraction of observations where a determined class of PSC occurs. The upper row displays the CALIOP PSC product, while the lower row shows the PSC classification obtained by applying the approximate algorithm to the ground-based data.

|  | 2006 | | July 2006 | | August 2006 | |
|---|---|---|---|---|---|---|
| PSC classes | CALIOP | gr.based | CALIOP | gr.based | CALIOP | gr.based |
| STS | 15.5 | 14.6 | 12.8 | 11.1 | 15.1 | 11.4 |
| NAT mixtures | 73.6 | 76.0 | 67.5 | 72.2 | 83.1 | 86.6 |
| enhanced NAT mixtures | 2.3 | 2.4 | 2.8 | 3.8 | 0.8 | 0.2 |
| ice | 8.5 | 7.1 | 16.9 | 12.9 | 1.0 | 1.8 |
| overpasses/observations | 128 | 75 | 35 | 31 | 37 | 22 |

**Table 2.** Frequency of occurrence (in %) of PSC classes for 2006 and for July and August separately. The last line represents the number of overpasses in the McMurdo box for CALIOP and the number of ground-based observations. Observations in the 12-30 km interval have been considered. The ice-class for CALIOP includes also mountain wave ice.

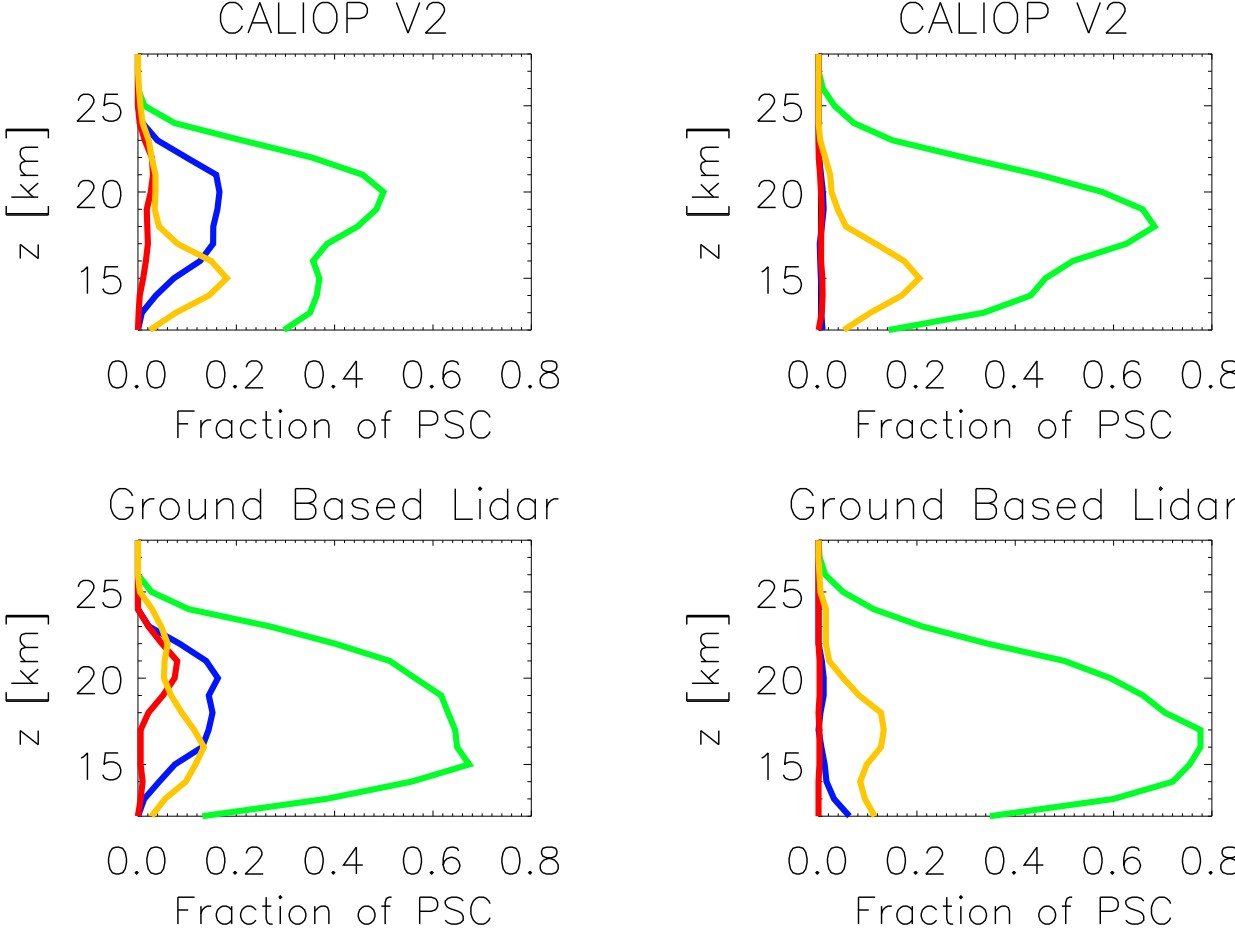

**Figure 3.** The PSC vertical distribution for the 2006 winter as a fraction of the total observations for the four PSC classes (orange = STS, green = NAT mixtures, red = enhanced NAT mixtures, blue = ice), the three columns indicate the months July and August (from left to right). Upper row: CALIOP v2 product. Lower row: ground-based lidar at McMurdo.

The figure shows that PSCs are observed up to 25 km in July and August. Above 25 km the number of PSC observations is negligible, both for ground-based and CALIOP observations. NAT mixtures are the dominating species with a slightly different altitude ditribution in July; ground-based occurrences of NAT mixtures are more frequent below 18 km with respect to CALIOP data.

The occurrences of ice clouds in July are very similar, while in August some low ice clouds appear in the ground-based data, but are absent in the CALIOP observations. Enhanced NAT mixtures occur mainly in July, and are observed between 17 and 25 km, though more abundant in the ground-based observations. The vertical distribution of STS shows a good agreement in July and August.

Another way to compare the statistical distribution of PSCs as observed by both instruments is to use the temperature dependence. The temperature dependence of the occurrence of different PSC classes has been studied intensively with in-situ and remote data with the goal to to confirm hypotheses on microphysical mechanisms of PSC formation (Peter, 1997). In this context we want to use it as another tool to investigate a possible bias when comparing ground-based and satellite based observations centered on McMurdo. The temperature data base used for the data analysis of CALIOP is MERRA-2 (Modern Era Retrospective analysis for Research and Applications) which uses the GEOS-5 analysis. In a previous analysis of the McMurdo ground-based lidar data (Di Liberto et al., 2014), the temperature was obtained from radiosoundings and, where these were not available, from NCEP. For the present analysis, however, we choose to use the same MERRA-2 temperature data for the ground-based data, in order to avoid a temperature bias while comparing with CALIOP data. The ice formation temperature $T_{NAT}$ has been obtained from daily values of the EOS MLS retrieved data for $HNO_3$ and $H_2O$ number densities.

The probability density functions of the different species are reported in figure 4 as the ratio of the occurrence of each species and the total number of observations at the specific temperature T-$T_{NAT}$. The total number of observations is reported as well, in arbitrary units, to indicate the variation of the number of observations with temperature. The ratio of occurrence has not been displayed when the number of observations at a specific temperature is too low to be statistically valid (less than 5 % of the total observations.

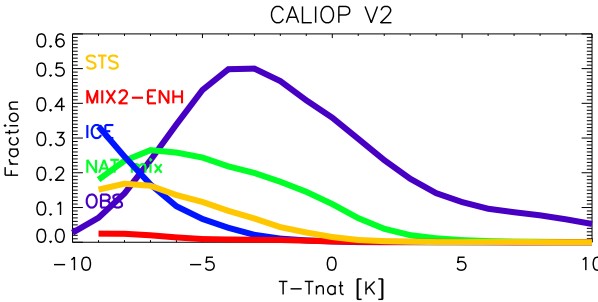 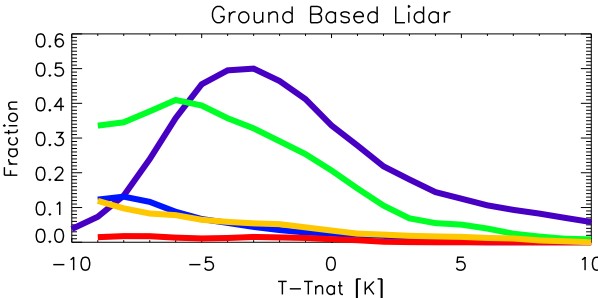

**Figure 4.** Fraction of PSC observations in 2006-2010 centered at McMurdo (calculated as the ratio of the number of data points for each PSC class and the total number of data points) as a function of the difference between the temperature and the equilibrium temperature for NAT. PSC classes are reported in different colors. The purple line indicates the total number of observations at a specific temperature in arbitrary units.

The total number of observations have a very similar temperature distribution, which indicates that the two instruments statistically sample air masses with a similar temperature distribution. The temperature dependence of the NAT and STS PSCs is very similar, although the peak for NAT is slightly shifted to lower temperatures. The onset for ice is the same, although the ice fraction at lower temperatures appears to be more larger for CALIOP with respect to the ground-based data. This is probably due to the fact that ice is not frequently observed around McMurdo (Adriani et al., 2004; Di Liberto et al., 2014) and that the few observations occur at different altitudes as can be seen also in figure 3.

## 3  Comparison of CALIOP PSC observations in the Southern Hemisphere with CCM simulations

The coupling of stratospheric chemical models with climate models has led to a new generation of models. These coupled CCMs have been used within the Chemistry-Climate Model Validation activity 2 (CCMVal-2) (Eyring et al., 2008) and represent both stratospheric chemistry and atmospheric climate. CCMVal-2 models do not include a representation of stratospheric aerosol physics and chemistry, but use parametrizations to take into account the formation of PSCs. There are large differences among CCMs for their treatments, regarding their formation mechanisms, types, and sizes (Morgenstern et al., 2010). All CCMs involved in the CCMVal-2 experiment include water-ice PSCs; all except CMAM also include nitric acid trihydrate (NAT). Most CCMs furthermore treat sulfate aerosols, e.g. in the form of supercooled ternary solutions (STS) of sulfuric acid ($H_2SO_4$), nitric acid ($HNO_3$), and water (Morgenstern et al., 2010).

Evaluating the ability of CCMs to reproduce ice and NAT PSCs is a key factor to interpret simulated stratospheric polar ozone changes. The comparison of space-borne PSC observations with CCM simulations requires adequate diagnostic methods. Here we assess the ability of models to simulate PSCs taking into account diagnostics that mostly focus on microphysical factors, such as the NAT and ice surface area densities and diagnostics that are sensitive to the coupling of those with the simulation of polar vortex variability and its mean state.

## 3.1 Overview of the models

Here we consider 4 CCMs involved in the CCMVal-2 experiment, CAM3.5 (Community Atmosphere Model 3.5) (Lamarque et al., 2008) and WACCM (Whole-Atmosphere Chemistry-Climate Model) (Garcia et al., 2007) both developed at NCAR, CCSRNIES (Center for Climate System Research/National Institute for Environmental Studies, Japan) (Akiyoshi et al., 2009), and LMDZrepro (Laboratoire de Météorologie Dynamique Zoom- REPROBUS) (Jourdain et al., 2008), developed at IPSL (Institut Pierre-Simon Laplace), and one CCM included in the Chemistry–Climate Model Initiative (CCMI), WACCM-CCMI (Solomon et al., 2015; Garcia et al., 2017).

Some general features such as the horizontal resolution and vertical levels have been displayed in Table 3.

| CCM | Years | Horizontal resolution | vertical grid | References |
|---|---|---|---|---|
| CAM3.5 | 1991-1999 | 2.5° x 1.9° | L26 | Lamarque et al. (2008) |
| CCSRNIES | 1991-2005 | 2.8° x 2.8° | L34 | Akiyoshi et al. (2009) |
| LMDZrepro | 1991-2005 | 3.75° x 2.5° | L50 | Jourdain et al. (2008) |
| WACCM | 1995-2005 | 2.5° x 1.9° | L66 | Garcia et al. (2007) |
| WACCM-CCMI | 1960-2010 | 2.5° x 1.9° | L66/88 | Solomon et al. (2015); Garcia et al. (2017) |

**Table 3.** Horizontal resolution and number of levels for the CCMs used. The output of the models has been taken for the years indicated in the second column.

All models include water-ice PSCs as well as NAT. They also treat sulfate aerosols in different forms, such as STS (CAM3.5, WACCM and CCSRNIES), or liquid aerosol (LMDZrepro).

The conditions at which PSCs condense and evaporate vary, not only for water-ice PSCs but also for NAT and STS, between CCMs (Morgenstern et al., 2010). Most CCMVal-2 models use a thermodynamic equilibrium assumption that PSCs are formed at the saturation points of $HNO_3$ over NAT and $H_2O$ over water-ice.

The microphysical processes of condensation and evaporation of the PSCs vary among the different models. CAM3.5 and WACCM allow for saturation of up to 10 times saturation (Morgenstern et al., 2010). Table 4 illustrates how the CCMs considered here use different formation processes and sedimentation velocities.

| CCM | Thermodynamics | particles | NAT/Ice Sedimentation |
|---|---|---|---|
| CAM3.5 | NAT: HY; ice:EQ | NAT/ice/STS | radius dependent |
| CCSRNIES | EQ | NAT/ice/STS | radius dependent |
| LMDZrepro | EQ | NAT/iice/LA | |
| WACCM | NAT: HY; ice:EQ | NAT/ice/STS | radius dependent |
| WACCM-CCMI | NAT: HY; ice:EQ | NAT/ice/STS | radius dependent |

**Table 4.** Main features of simulation and of the microphysics of polar stratospheric clouds. EQ =thermodynamic equilibrium with gaseous $HNO_3$ / $H_2SO_4$ / $H_2O$ assumed. HY = non-equilibrium / hysteresis considered. LA=liquid aerosol (adapted from CCMVal-2 report (2010)).

Note that the equilibrium assumption allows to determine the total mass of condensed PSCs, and that a size distribution needs to be postulated in order to derive surface area densities (SAD). Since the sedimentation velocity depends on the size of the particles, the size distribution assumed has a significant impact on denitrification and dehydration processes through sedimentation of PSCs.

Some differences between WACCM and WACCM-CCMI should be mentioned here. While the CCMVal-2 version of WACCM simulated Southern Hemisphere winter and spring temperatures that were too cold compared with observations, in the CCMI-1 simulations this problem was addressed by introducing additional mechanical forcing of the circulation via parametrized gravity waves(Garcia et al., 2017). Also the polar heterogeneous chemistry was recently updated (Wegner et al., 2013) and further evaluated by Solomon et al. (2015).

Recently Zhu and co-workers introduced a new PSC model (Zhu et al., 2015, 2017a, b) within the CESM1 (Community Earth System Model) Whole Atmosphere Community Climate Model version 4.0 (WACCM 4.0), with Specified Dynamics (SD) coupled with the Community Aerosol and Radiation Model for Atmospheres (CARMA) model. This new model takes into account detailed microphysical processes for the formation of NAT and STS, instead of the parametrizations used in the CCMVal-2 and CCMI-1 models. An evaluation study on the ECHAM5/MESSy Atmosperic Chemistry (EMAC) model has been reported (Khosrawi et al., 2018), using MSBM (multi-phase stratospheric box model) for the processes related to PSCs (Kirner et al., 2011). The submodel MSBM uses two parametrizations for the NAT formation, one based on the heterogeneous formation on ice, the second for the homogeneous formation of NAT. The model simulations for the Arctic winter 2009/2010 and 2010/2011 showed that simulated PSC volumes are smaller than those observed and that the simulations do not produce PSCs as high as they are observed.

These models are, to our knowledge, the most significant advancements in the field of PSC representation in Global Climate Models used for ozone and climate change studies. The CARMA model is an interactive aerosol and radiation model fully coupled to the WACCM, able to simulate advection, diffusion, sedimentation, deposition, coagulation, nucleation and condensational growth of atmospheric aerosols online with the temperature, dynamics and radiation structure simulated by the GCM. This approach is completely different from the parametrizations available in the simulations we are analysing here. A

full evaluation of the WACCM/CARMA models in Specified Dynamics runs with respect to CALIPSO data is available in literature (Zhu et al., 2015, 2017a, b) but is beyond the scope of this intercomparison, where free running simulations are used.

Here we limit our analysis to simulations produced by four models from CCMVal-2 and one model from CCMI. One of the goals is to use different diagnostics to test the model simulations versus the CALIOP observations. Recent studies of model simulations of PSCs can be found in (Brakebusch et al., 2013; Wegner et al., 2013; Zhu et al., 2015, 2017a).

## 3.2  Comparison based on the PSC vertical extent

Presently, the evaluation of CCMs for what concerns stratospheric aerosol and in particular PSCs is still incomplete. The SPARC report (Eyring et al., 2010) includes a model inter-comparison of PSC surface area densities (SAD) concluding that more work is needed to evaluate NAT and ice aerosols and that a comparison with observations is clearly needed, since currently no global data sets are available to evaluate these constituents. The CCMVal-2 data base includes the surface area density (SAD) for NAT and ice clouds and sulphates. Here we restrict the analysis to the reference simulations (REF-B1), the transient set of simulations aiming at reproducing the past 1960-2006 conditions where all forcings are taken from observations (Eyring et al., 2010).)

Both model and CALIPSO observations in the latitude range [82°,60°] are binned in a 3.5°x7° grid and on 15 vertical levels with a resolution of 1.5 km. CCMVal-2 data south of 82°S are excluded to fit with CALIPSO latitudinal coverage. In this study, we use two different WACCM versions, with the same PSC scheme, used within CCMVal-2 and CCMI.

To be able to compare with the CALIOP lidar observations, we have to derive the mean PSC layer vertical extent and the frequency of occurrence as a function of height and of temperature for the models from the PSC surface area density (SAD) spatial distribution. To do so, it is necessary to apply a simplified observation operator to the model output (i.e. identify the model grid points where a lidar would have observed NAT or ice clouds by defining a threshold for the SAD values produced by the models). We firstly define a vertical extent of PSCs as the sum of all layers (in km) containing a specific class of PSC. In order to study seasonal and geographical variations, we construct maps of monthly means by accumulating all observations.

Figure 5 shows maps of the monthly mean vertical extent (in km) for ice, NAT mixtures, enhanced NAT mixtures and STS PSCs as observed by CALIOP from 2006 to 2010.

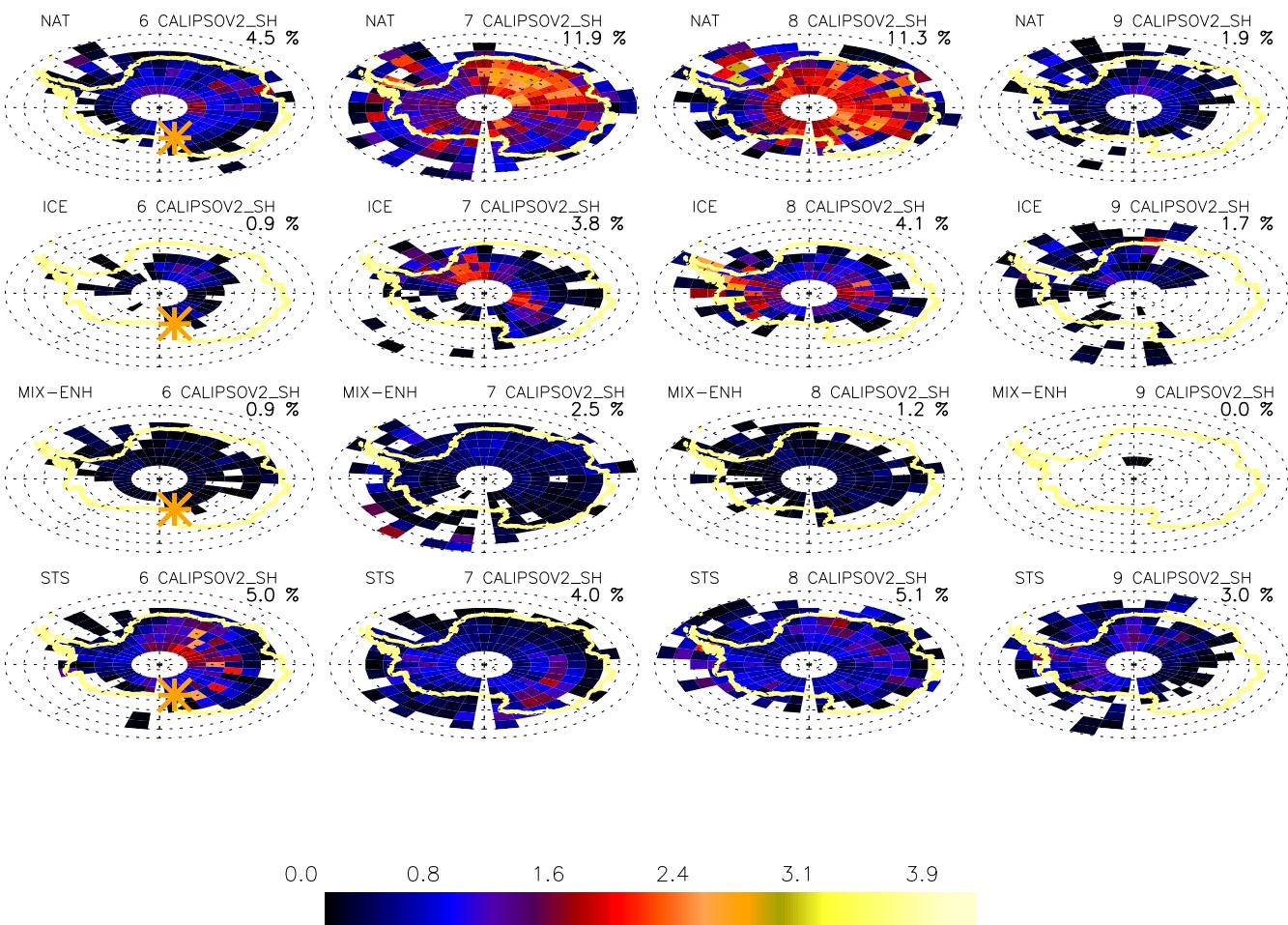

**Figure 5.** The vertical extent for NAT mixtures, ice, enhanced NAT mixtures and STS (from top to bottom) obtained from CALIOP observations, averaged over 5 years for June, July, August, September (left to right) is displayed. In the left column the location of McMurdo is indicated with a yellow asterisk. The fraction of the overall air volume (between 12 and 30 km height south of 60°S) occupied by different PSC classes for each month is reported in the top right corner. The colour scale indicates the number of km occupied by PSCs between 12 and 30 km.

The ice PSC distribution has a clear non-zonal longitudinal distribution with a maximum in the 90° W - 0° longitude sector. This appears as an indication that mountain waves play a major role in ice cloud formation on the lee side of the Transantarctic chain, crossing the continent as an ideal prolongation of the Antarctic Peninsula. This has previously been reported by Noel et al. (2009) and by Alexander et al. (2011) based on the combination of CALIPSO and COSMIC (Constellation Observing System for Meteorology, Ionosphere, and Climate) GPS-RO (Global Positioning System Radio Occultation) data. The latter reports an analysis based on a single winter data set showing that mountain wave generation is a regular feature influencing ice

PSC distribution. NAT-like (NAT plus enhanced NAT) PSCs have a maximum in the 0° - 90°E longitude quadrant. Höpfner
10    et al. (2006) suggested that mountain waves may be responsible for the non-zonal NAT distribution that were indeed observed
closer to the Transantarctic chain while Alexander et al. (2011) also consider that NAT formation can be related to the outflow of
ice clouds. Wang et al. (2008) pointed out that increased convection due to orographic triggering in the lee of the Transantarctic
chain is related to the occurrence of enhanced NAT mixtures. Enhanced NAT mixtures have a minor vertical extent with respect
to NAT mixtures and form in the inner vortex (where colder temperatures occur) with a zonal distribution similar to NAT
mixtures. STS are observed predominantly in June, again with a clear majority in the same region of NAT and enhanced NAT
mixtures formation. The McMurdo site is characterized by a majority of NAT-like PSCs (also visible in the time-series reported
in figure 2).

Here we compare maps of NAT and ice PSC occurrences, produced by the five models, showing the geographical distribution
5     of NAT and ice in the southern hemisphere (south of 60°S) for the winter season, from June to September (see figures 6, 7, 8
9 and 10) with CALIOP observations (figure 5).

The vertical extent for the models is estimated analogously to the observations. The horizontal resolution applied to estimate
the occurrence is the same among models and CALIOP data. The effect of the differences of vertical resolution among models
and observations is reduced by calculating a total aggregate vertical occurrence.

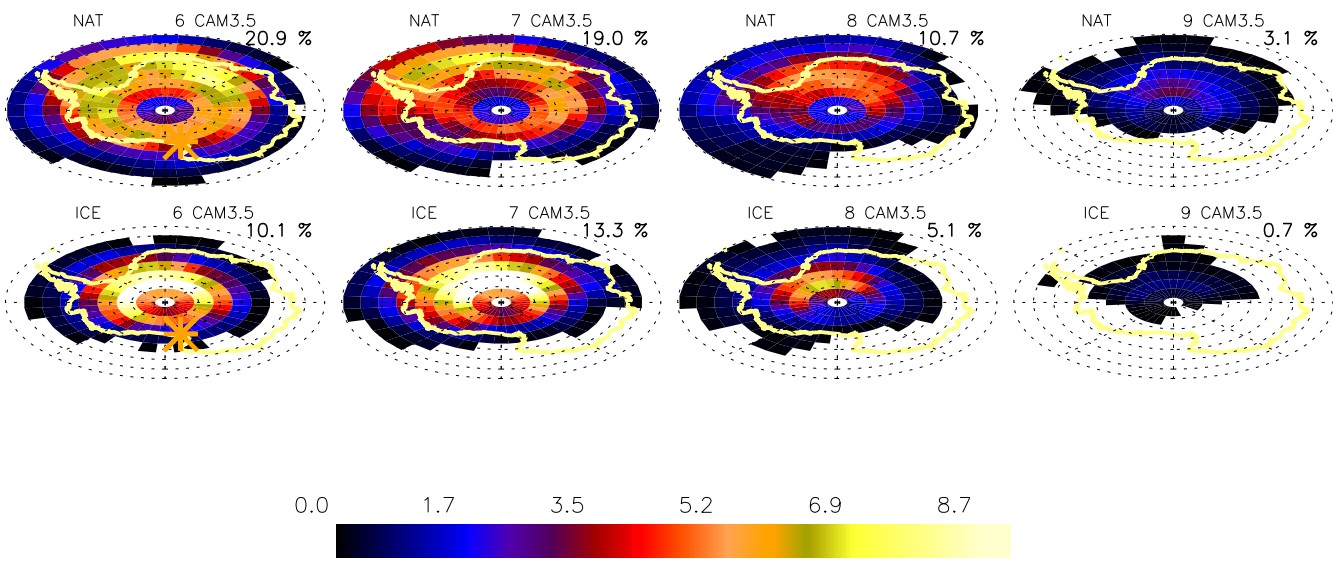

**Figure 6.** CAM3.5 PSCs vertical extent for NAT and ice, averaged over five years during the months of June, July, August, September (left
to right). Please note that the color scale is different from the other maps. In the left column the location of McMurdo is indicated with a
yellow asterisk. The fraction of the overall air volume (between 12 and 30 km height south of 60°S) occupied by different PSC classes for
each month is reported in the top right corner.

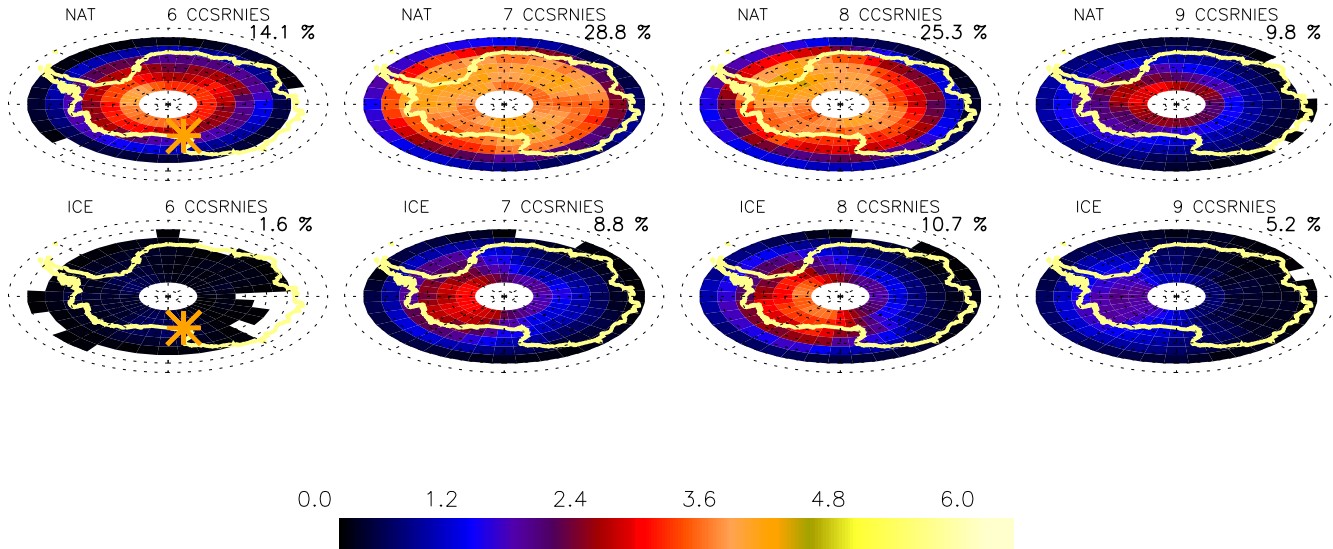

**Figure 7.** CCSRNIES PSCs vertical extent for NAT and ice, averaged over five years during the months of June, July, August, September (left to right). Please note that the color scale is different from the other maps. In the left column the location of McMurdo is indicated with a yellow asterisk. The fraction of the overall air volume (between 12 and 30 km height south of $60°$S) occupied by different PSC classes for each month is reported in the top right corner.

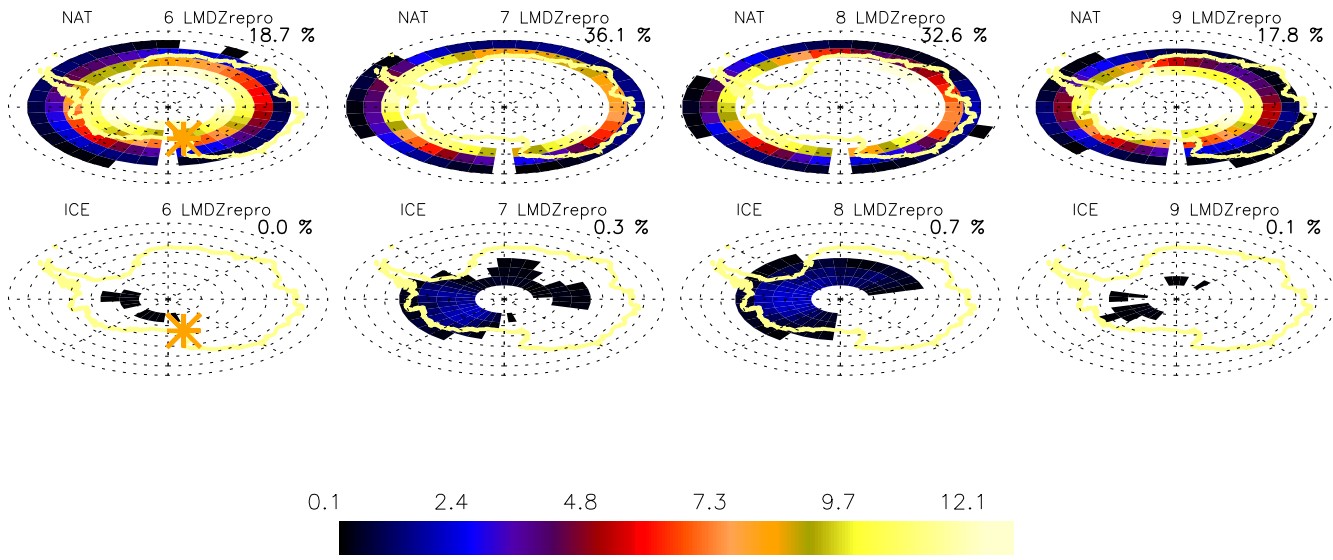

**Figure 8.** LMDZrepro PSCs vertical extent for NAT and ice, averaged over five years during the months of June, July, August, September (left to right). Please note that the color scale is different from the other maps. In the left column the location of McMurdo is indicated with a yellow asterisk. The fraction of the overall air volume (between 12 and 30 km height south of $60°$S) occupied by different PSC classes for each month is reported in the top right corner.

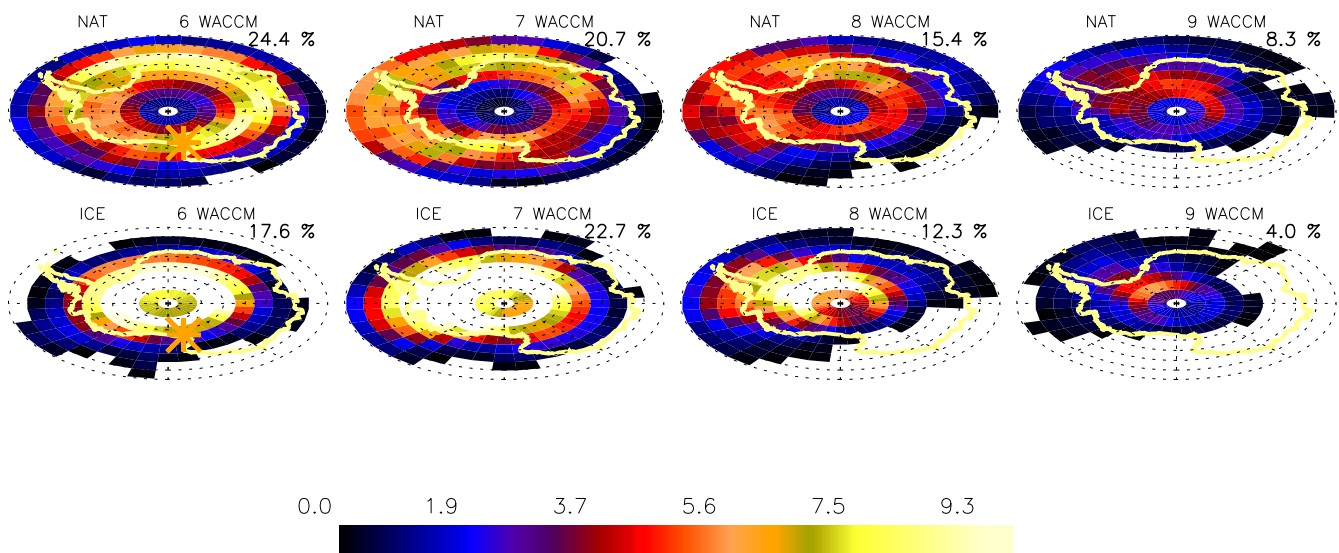

**Figure 9.** WACCM PSCs vertical extent for NAT and ice, averaged over five years during the months of June, July, August, September (left to right). Please note that the color scale is different from the other maps. In the left column the location of McMurdo is indicated with a yellow asterisk. The fraction of the overall air volume (between 12 and 30 km height south of 60°S) occupied by different PSC classes for each month is reported in the top right corner.

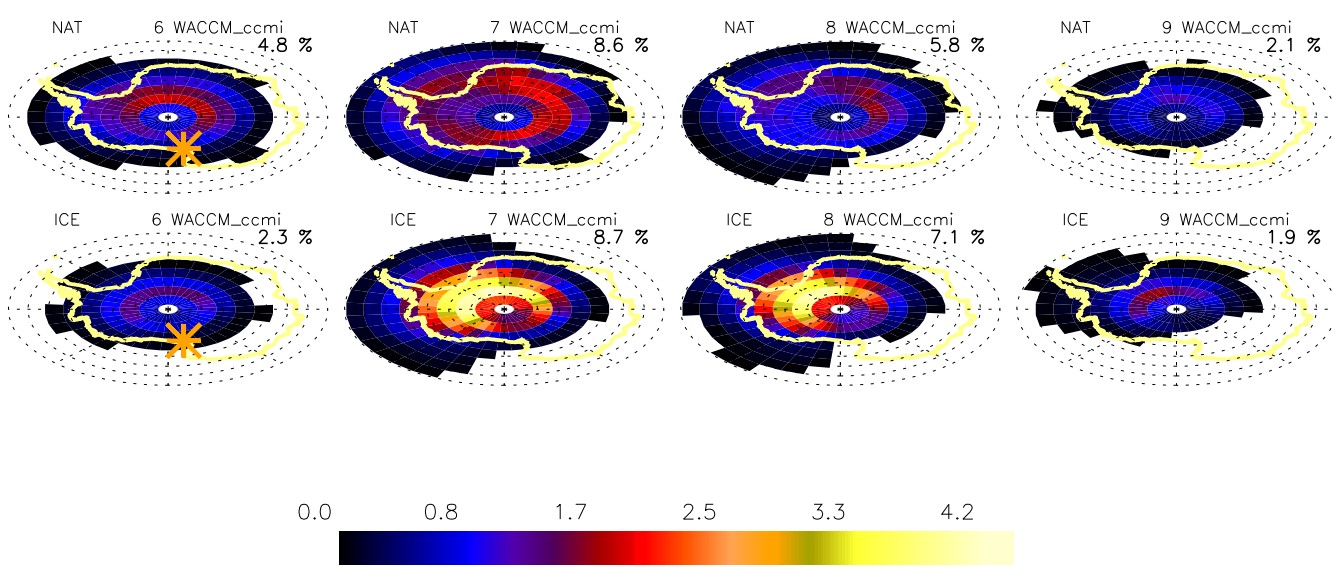

**Figure 10.** WACCM-CCMI PSCs vertical extent for NAT and ice, averaged over five years during the months of June, July, August, September (left to right). Please note that the color scale is different from the other maps. In the left column the location of McMurdo is indicated with a yellow asterisk. The fraction of the overall air volume (between 12 and 30 km height south of 60°S) occupied by different PSC classes for each month is reported in the top right corner.

     Table 5 reports the total PSC vertically integrated frequencies of occurrence for the five models and for CALIPSO from June to September as already indicated in figures 5,6, 7, 8, 9 and 10.

| | NAT mixtures | | | | ice | | | |
|---|---|---|---|---|---|---|---|---|
| | Jun | July | Aug | Sep | Jun | July | Aug | Sept |
| CALIPSO | 4.5 | 11.9 | 11.3 | 1.9 | 0.9 | 3.8 | 4.1 | 1.7 |
| CAM3.5 | 20.9 | 19.0 | 10.7 | 3.1 | 10.1 | 13.3 | 5.1 | 0.7 |
| CCSRNIES | 14.1 | 28.8 | 25.3 | 9.8 | 1.6 | 8.8 | 10.7 | 5.2 |
| LMDZrepro | 18.7 | 36.1 | 32.6 | 17.8 | 0.0 | 0.3 | 0.7 | 0.1 |
| WACCM | 24.4 | 20.7 | 15.4 | 8.3 | 17.6 | 22.7 | 12.3 | 4.0 |
| WACCM-CCMI | 4.8 | 8.6 | 5.8 | 2.1 | 2.3 | 8.7 | 7.1 | 1.9 |

**Table 5.** Total PSC frequencies (in %) in the 13-25 km height layer for NAT and ice clouds for June-July- August-September for the observations and models. Fractions below 1% are not reported in the table. Note that CALIPSO NAT includes the enhanced NAT mixtures class

The differences between the simulations obtained from the CCMs and CALIOP observations are discussed in terms of geographical distribution, onset and decline of PSCs during polar winter and total vertical extent for NAT and ice.

The CAM3.5 model overestimates NAT and ice throughout the winter and shows an early onset of PSCs in June and also an early decline in August, with respect to CALIOP observations.

Also CCSRNIES shows a too strong presence of NAT and ice, with respect to CALIOP, in particular in September, but shows a correct seasonality, with July and August being the months with the largest presence of PSCs.

The LMDZrepro model produces a correct onset and decline of the PSC formation, but shows the largest NAT frequency and the lowest ice frequency of all models.

WACCM is similar to CAM3.5, but with a larger NAT and ice frequency. The onset of PSC formation is early, as for CAM3.5.

The simulations produced by WACCM-CMMI follow the same trend for both NAT and ice, as observed by CALIOP, although the NAT frequency in July ad August is underestimated and the ice PSCs are overestimated in July, August and September, with respect to CALIOP.

In discussing the geographical distribution of PSCs, it should be noticed that the small numbers of observations in some cases makes any comparisons among models and CALIOP difficult. The NAT occurrences as observed by CALIOP in July and August are mainly concentrated in East Antarctica, while ice is more manifest towards the 90°W direction.

The geographical distribution of NAT and ice in the CAM3.5, WACCM and WACCM-CMMI simulations is similar, with a dominant presence in the 90°W - 0° sector. CCSRNIES shows a more symmetric distribution of NAT, with a slight increase in

10    the 90°W - 0° sector in particular in July and August. LMDZrepro shows a very strong presence of NAT, with a preference of the 90°W - 0° sector, while ice is present in very small amounts, mostly around 90°W.

As a conclusion the WACCM-CCMI model compares better with observations, for what concerns the seasonal behaviour and the occurrences. The reduction of the cold bias in the WACCM-CCMI version (Doug Kinnison personal communication) may be the most relevant factor leading to a better agreement with observations with respect to the older versions of the model

15    (WACCM and CAM3.5).

All other models overestimate the NAT occurrences, most probably due to the cold temperature bias. Also ice is much overestimated, with the exception of LMDZ-repro which underestimates the ice occurrences with respect to CALIOP.

### 3.3   Comparison based on SAD

Another diagnostic method consists of comparing the SAD for CCMs and CALIOP. A range of SAD values can be obtained for NAT and ice for each model. The surface area density for the CCMVal-2 is estimated based on a semi-empirical relation between mass and mean surface areas given by the model providers and reported in the CCMVal-2 report. We must be aware, however, that SAD is a derived variable and depends on the assumptions on the mean particle size for each model (as detailed

5    in the CCMVal-2 report, 2010). When models predict both NAT and ice clouds, we assigned the SAD to ice if the SAD for ice is larger by a factor of 3 than the one for NAT. The SADs for CALIOP have been evaluated by using an empirical relationship derived from coincident lidar and size distributions observations (Snels et al., 2018). Figure 11 shows the histograms of ice and NAT values for SAD for each model together with the range of SAD reported in Adriani et al. (1995). The fraction is normalized to the total number of model grid points in order to identify the differences in PSC occurrence among models and between classes.

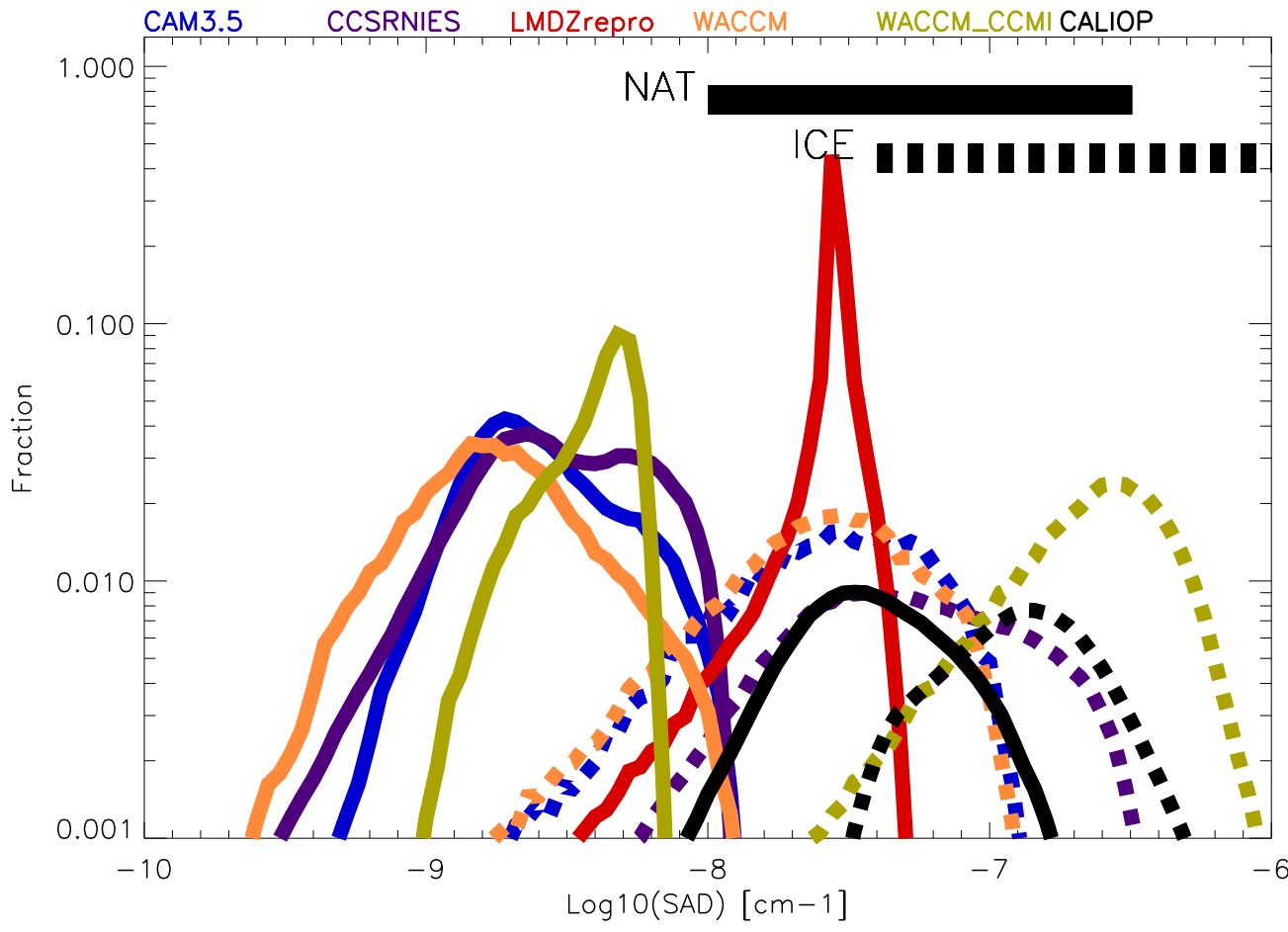

**Figure 11.** Histogram of the NAT (solid lines) and ice (dashed lines) SADs for some CCMVal models and for CALIOP are displayed. The histograms for the model data have been truncated and represent 93% of the total SAD. The straight lines at the top of the figure indicate the range of SAD values for NAT and ice "observed" by ground-based lidars and are taken from Adriani et al. (1995).

We observe that for most of the models NAT PSCs have SAD ranging between $3 \cdot 10^{-10}$ and $10^{-8}$ cm$^{-1}$ except for LMDZre-pro that has larger SAD for NAT PSCs and is clearly an outlier. In general all models produce SADs for NAT that are smaller by one order of magnitude than the SAD calculated from CALIOP data, except for LMDZ-repro. The variability among models for the NAT SAD may be related to the assumptions made on the number of particles per cm$^{-3}$. The narrow peak at larger NAT SAD values for the LMDz model could be consistent with the use of much larger particle number density and smaller particle radius in the simulation. This in turn would give less irreversible denitrification processes simulated by the models with larger NAT SAD (CCMVal-2 report, 2010, Chapter 6). Most of the models have ice PSCs in a SAD range between $2 \cdot 10^{-9}$ and $10^{-6}$ cm$^{-1}$ and are generally a factor of 2-3 smaller than CALIOP values, except for the WACCM-CCMI simulations, which predict a larger value than that derived from CALIOP observations.

## 3.4 Comparison based on PSC occurrences

The comparison between CALIOP and CCMs can also be made by using the occurrences as a function of T-T$_{NAT}$, similarly to what has been done above for the comparison between ground-based and satellite-borne lidars above McMurdo. In figure 12 the PSC occurrences as predicted by the models and as observed by CALIPSO between 60°S and 82°S averaged over the 2006-2010 period have been displayed as a function of T-T$_{NAT}$, where T$_{NAT}$ has been calculated from HNO$_3$ and H$_2$O number densities. Note that the models produce only NAT and ice occurrences.

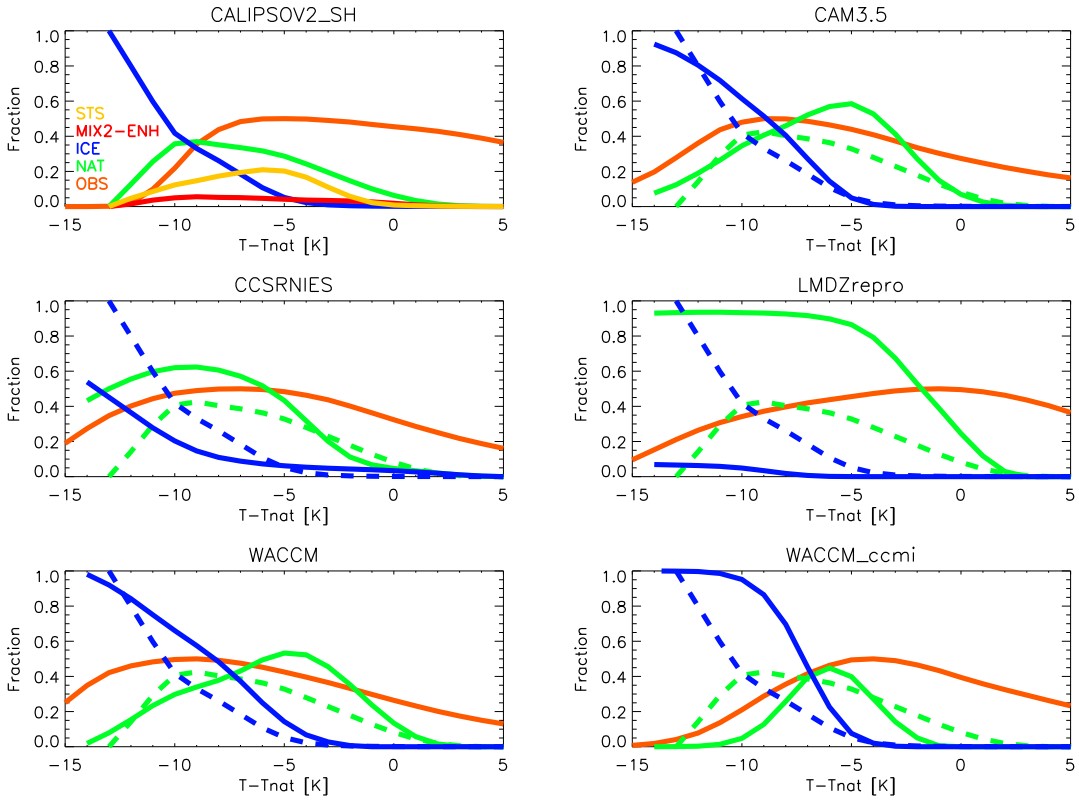

**Figure 12.** As Figure 4 but for 60-82°S CALIOP v2 observations and CCMs data. CALIOP data are reported as dashed lines on each model as reference. The orange curve (OBS) is a histogram for the temperature distribution of all observations (CALIOP) or simulations (models).

As reported in the CCMVal-2 report, most models show a well-known cold pole bias in stratospheric temperature. The bias is in general attributed to model dynamics, as in (Austin et al., 2003) that identifies a lack of westward wave forcing resulting in a more intense and persistent polar vortex. A clear improvement is obtained with an improvement in the gravity waves scheme as in (Kinnison et al., 2007), resulting in more realistic temperatures in the WACCM-CCMI simulation as described above.

    The fraction of data with different PSC classes helps in evaluating how realistic the microphysical scheme is, since this
variable is normalized to the number of observations and in principle independent from the possible biases. The onset of

NAT is similar for all models, except for WACCM-CCMI, where NAT starts to form only below $T_{NAT}$. The onset of the ice formation occurs at $T-T_{NAT} = -5$ K for all models, except for CCSRNIES. The increase of NAT occurrences with decreasing temperatures is stronger for all models with respect to CALIOP. This is due to the fact that the models consider only the thermodynamic equilibrium conditions for the formation of PSC, and do not allow the existence of supersaturation without

PSC formation. The family of models CAM3.5, WACCM and WACCM-CCMI show a faster increase of the ice occurrences with decreasing temperatures with respect to CALIOP. The reason is probably the same as for the NAT behaviour. LMDZ-repro evidently produces much less ice than the other models and CALIOP, and at low temperature NAT is the dominating species, while the other models and CALIOP show a dominant ice occurrence for low temperatures. The CCSRNIES model shows a slower increase of the ice occurrences with respect to CALIOP and the other models.

In general CAM3.5 and WACCM that share the same microphysical scheme have a more than satisfactory agreement, notwithstanding the cold bias that generates an excessive PSC coverage. On the other hand, WACCM-CCMI has a more realistic PSC coverage but a likely too efficient ice PSC generation due to the new scheme. So, even if the overall skills of the model are largely improved, this kind of diagnostics (the slopes of curves in figure 12 and the "onset" PSC temperature) suggest the need to explore the ability of a single component of the model system such as the microphysical scheme.

## 25   4   Conclusions

A statistical comparison has been proposed for PSC observations at McMurdo, obtained from ground-based and satellite-borne lidar measurements. The analysis of the ground-based data has been performed by using a detection and classification algorithm which closely follows the v2 algorithm applied to CALIOP data, in order to avoid a bias due to different classification schemes. Results have been shown for July and August 2006, being the months with the best temporal coverage. A comparison of PSC

occurrences as a function of time and height in 2006, shows that both datasets capture the general features of the PSC season, in terms of occurrence of each species throughout the winter. The vertical distribution and the temperature dependence of the occurrences of the different PSC classes show some discrepancies, in particular there are noticeable differences in the height distribution of NAT around 20 km. As a conclusion, the statistical agreement between CALIOP and ground-based data is acceptable, considering the different observation geometry and other possible biases.

A set of diagnostics has been proposed to compare the PSC simulations from CCMs with respect to CALIOP, with the goal to evaluate possible biases. The diagnostics are based on spatial (vertical and horizontal) SAD distribution of ice and NAT particles together with their temperature distributions. Those diagnostics are here applied to a subset of CCM simulations from

5 CCMVal-2 and to a more recent version of WACCM from CCMI. The geographical distributions of PSCs in the polar vortex observed by CALIOP is not well reproduced by most of the models. Moreover the NAT frequency is overestimated, with respect to CALIOP for all models, except for WACCM-CCMI. The onset of PSC formation is anticipated in the CAM3.5 and WACCM models, with respect to CALIOP, while CCRSNIES and LMDZrepro show a too strong presence of NAT in June and September with respect to July and August. LMDZrepro has the largest amount of NAT and the smallest amount of ice PSCs. WACCM-CCMI shows the best agreement with CALIOP, both for onset and decline and for absolute values, although NAT is

slightly underestimated in July and August and ice is overestimated in the same months. As a conclusion the WACCM-CCMI model compares better with CALIOP observations for ice and NAT, due to additional forcings applied in order to eliminate the cold temperature bias.

*Acknowledgements.* The authors acknowledge the financial support by PNRA in the frame work of the projects 2004/2.09 and 2009/B.08. We also acknowledge the support of the ISSI-PSC initiative project. Logistical and winter-time technical support was provided by the US

National Science Foundation through NSF awards 0538679 and 0839124 to the University of Wyoming.

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
