# Peer review of "Comparison of Antarctic polar stratospheric cloud observations by ground-based and spaceborne lidars and relevance for Chemistry Climate Models"

_Atmospheric Chemistry and Physics, 2018_

## Referee Comment (RC1) · Anonymous Referee #1 · 18 Sep 2018

A Review of "Comparison of Antarctic polar stratospheric cloud observations by ground-based and spaceborne lidars and relevance for Chemistry Climate Models" by M. Snels et al.

<General Comments>

This paper describes the comparison between PSC measurements at Antarctic Mc-Murdo Station from ground based lidar and CALIOP satellite measurements. Furthermore, the paper tries to extend the comparison of PSC statistics from CALIOP with

several CCM model results from CCMVal-2 and CCMI. Although scientific value of this study might be significant, the method of comparison especially with CCM models is not well organized to derive scientifically useful conclusions, as is pointed out below. Also, there are too many typos and careful mistakes in the draft. A major revision is required before this paper will be published in ACP. I recommend that authors should check the draft carefully, including the native check, before submitting the revised draft.

<Major Comments>

(M1) In Section 3.2, the authors try to compare the PSC statistics from 5 years (2006-2010) measurements by CALIOP, with the result of 4 CCM models from CCMVal-2, and one CCM model from CCMI. However, the model run type they chose for CCMVal-2 models are REF-B2, which are targeted to be used for future predictions until 2100. The major problem for this comparison is that the result of REF-B2 run contains both inaccuracy in modeled temperatures and imperfectness in PSC schemes which are different in each model. The combination of inaccuracies both in modeled temperature and PSC schemes makes it extremely difficult to understand the nature of PSC in each model. Rather than comparison with CCMVal-2 REF-B2 runs, it is strongly preferred to compare with CCMI outputs with refC1SD runs (which is available from http://badc.nerc.ac.uk/browse/badc/wcrp-ccmi/data/CCMI-1/output), which use nudging with more realistic temperature and wind field, just to test the PSC scheme in each model. Even if the authors stick to the comparison with CCMVal-2 model results, they should at least use the REF-B1 model run results, which are targeted to reproduce the past. In this case, the comparison with CALIOP could be made only for 2006, because REF-B1 run was made only for 1960-2006. Since CCMI refC1SD runs cover until 2010, I strongly recommend making comparisons with CCMI model outputs with CALIPSO measurements.

(M2) In Section 3.1, the authors mention about more sophisticated WACCM4.0/SD/CARMA model and EMAC/MSBM model, which use more realistic parameterizations for PSCs. It would gain the value of this paper significantly if

they could include the comparison of CALIOP PSC statistics with the result of these models.

(M3) In each model, denitrification and dehydration are included as is shown in Table 3. This would change the vertical distribution of $HNO_3$ and $H_2O$, which would affect the threshold temperature of NAT and ice PSCs, i.e., $T\_NAT$ and $T\_ice$. However, this effect is never mentioned or discussed in the manuscript. Moreover, in many places in the text (especially in Sections 2.6 and 3.4), it is not clearly stated which temperature (MERRA-2, NCEP, or derived T in CCM) is used, and how $T\_NAT$ and $T\_ice$ are calculated (using $HNO_3$ and $H_2O$ value from MLS data, modeled value in CCM, or fixed values like 6 ppbv $HNO_3$ and 4.5 ppmv $H_2O$). The effect of denitrification/dehydration in modeled PSC should be discussed in the manuscript.

(M4) For a PSC classes comparison described in Table 1, although the percentage of each PSC class is similar, this does not prove that each one to one PSC is simultaneously observed both by ground-based lidar and by CALIOP. I would recommend authors to add the statistics showing one to one correspondence of comparison of PSC classes observed by tables like the attached tables. Table A shows the statistics when CALIOP measured specific class of PSC, what PSC was observed by McMurdo ground-based lidar, or no PSC was observed. Table B shows the statistics when ground-based lidar measured specific class of PSC, what PSC was observed by CALIOP, or no PSC was observed.

(M5) In Section 3.4, they discuss about the cold pole bias in most CCMVal-2 CCM models. However, when I read the SPARC report No.5 Chapter 4 "Section 4.3.5 Polar stratospheric cloud threshold temperatures" in page 128, there is an explanation that CCM models have warm bias and $A\_NAT$ and $A\_ice$ show low value compared with ERA-40 temperature. This description totally contradicts with the discussion described in Section 3.4. Please explain why such contradiction occurs.

\<Specific Comments\>

(S1) The numbers in author list are not ordered correctly, i.e., 1, 5, 2, 3, 4. It should be 1, 2, 3, 4, 5.

(S2) P1, L3: The abbreviation of CALIOP should be shown also in the abstract.

(S3) P1, L9: The meaning of "... and a selection simulations obtained ..." is unclear.

(S4) P1, L4: In Pitts et al. (2018, ACP), they use "v2" instead of "V2". Please check if V2 should be changed to v2 throughout the manuscript or not.

(S5) P1, L18: The abbreviation of WACCM-CCMI should be shown.

(S6) P2, L7: The abbreviation of CALIOP should be shown here, not at P2, L20.

(S7) P2, L18: Chemistry Climate Models –> Chemistry Climate Models (CCMs)

(S8) P2, L20: clouds and aerosol –> clouds and aerosols

(S9) P2, L26: Chemistry Climate Models –> CCMs

(S10) P2, L29: The SPARC Report No5 (2010) cannot be found in the reference list.

(S11) P2, L30: Chemistry Climate Models –> CCMs

(S12) P3, L1: Chemistry Climate Models (CCM) –> CCMs

(S13) P3, L14: CALIOP (Cloud Aerosol Lidar with Orthogonal Polarization) –> CALIOP

(S14) P3, L14: Details on CALIOP –> Details of CALIOP

(S15) P4, L16: Reference (Cairo et al., 1999) should appear at the end of Line 18.

(S16) P5, L14: CALIPSO V2.0 data –> CALIPSO v2 data

(S17) P5, L15: V2.0 –> v2

(S18) P5, L17: V1.0 and V2.0 –> v1 and v2

(S19) P5, L26: Version 1.0 –> v1

(S20) P6, L22: "We used monthly averages" for what? Explain.

(S21) P6, L24-25: The English meaning of the following sentence is unclear: "This procedure approaches the V2 algorithm applied to the CALIOP data as good as possible"

(S22) P7, L1: close to the edge –> close to the polar vortex edge

(S23) P7, L10: lidar observes from distances –> lidar observed at distances

(S24) P10, L9: What is "lower altitudes wrt"?

(S25) P11, L1: Change paragraph before "Another way ..."

(S26) P11, L21-22: I cannot understand the explanation of the last sentence why the distribution of ice is different between ground-based lidar and CALIPSO.

(S27) P12, L6: Chemistry-Climate Models (CCMs) –> CCMs

(S28) P12, L6: [Eyring et al., 2008] –> (Eyring et al., 2008); this reference cannot be found in the reference list.

(S29) P12, L19-22: Please show references of each CCMs here as well as in Table 2.

(S30) P12, L22: The abbreviation of IPSL should be shown.

(S31) P13, Table 2: I assume that model run years for CCMs in CCMVal-2 for REF-B1 is 1960-2006, and for REF-B2 is 1960-2100. What do the years in this table mean?

(S32) P13, L1: nitric-acid-trihydrate (NAT) –> NAT

(S33) P13, L2: supercooled ternary solutions of sulfuric acid, water and nitric acid (STS) –> STS

(S34) P13, Table 3: I think themodynamics of CAM3.5 is NAT:HY; ice:EQ, not EQ. Please confirm.

(S35) P13, Table 3: I think sedimentation of CAM3.5 is dep. On mode radius? Please confirm.

(S36) P13, Table 3: What is the sedimentation of LMDZrepro? Please show.

(S37) P13, Table 3 caption: SAD does not appear in the table, so this is obsolete.

(S38) P14, L8: The abbreviation of CESM1 should be shown.

(S39) P14, L12: The abbreviation of MSBM should be shown.

(S40) P14, L14-16: A reference for the last sentence should be shown.

(S41) P15, L8: The abbreviation of COSMIC GPS-RO should be shown.

(S42) P15, L9: McMurdo site: please show where is the McMurdo Station in Figure 5.

(S43) P18, L3: CCMVal report (2010) –> CCMVal-2 report (2010): This reference is missing in the reference list.

(S44) P18, L3: maximum often occurring in June –> maximum often occurring in July (?)

(S45) P18, L7: surface area density (SAD) –> SAD

(S46) P19, Figure 7 caption: Surface Area Densities –> SADs

(S47) P21, 14: What is the meaning of "overall skills f the"?

<Technical Corrections>

(T1) P1, L10: Period after "Models" is obsolete.

(T2) P2, L7-8: McMurdo station –> McMurdo Station

(T3) P3, L17: in (Hunt et al., 2009; Winker et al., 2009). –> in Hunt et al. (2009) and Winker et al. (2009).

(T4) P13, Table 3: WACCM-CMMI –> WACCM-CCMI

(T5) P13, Table 3: HNO_3 / H_2SO_4 / H_2O (subscript)

(T6) P14, L7: by (Solomon et al., 2015). –> by Solomon et al. (2015).

(T7) P14, L8: (Zhu et al., 2017b, a, 2015) –> (Zhu et al., 2015, 2017a, b)

(T8) P14, L12: PSCS –> PSCs

(T9) P14, L17: CCM1 –> CCMI

(T10) P15, L5: 90° − 0° –> 90° W - 0°

(T11) P16, L2: 0° 90° –> 0°- 90°E

(T12) P17, L6: CMMI –> CCMI

(T13) P17, L7: WACCM-CMMI –> WACCM-CCMI

(T14) P18, L3: LMDZ –> LMDZrepro

(T15) P18, L11: CCMVAl-2 –> CCMVal-2

(T16) P18, L14: reported in (Adriani et al., 1995). –> reported in Adriani et al. (1995).

(T17) P19, Figure 7 caption: taken from (Adriani et al., 1995). –> taken from Adriani et al. (1995).

(T18) P19, L7: CCMVAl-2 –> CCMVal-2

(T19) P21, L1: WACCM-cmmi –> WACCM-CCMI

(T20) P21, L11: the for other –> the four other

(T21) P21, L13: notwithstanding –> not withstanding

(T22) P21, L31: CCMVal2 –> CCMVal-2

(T23) P23-26: Title of each word in the following references should be spelled in lower case letters: Eyring et al. 2010; Garcia et al., 2017; Hunt et al., 2009; Winker et al., 2009.

(T24) Reference list: The order of the following reference should be re-ordered to follow the order of published years: Pitts et al., 2013; 2018; 2009; 2011: Stephens et al., 2017; 2002.

(T25) P25, L8: Pitts et al. (2018) is now in ACP, not in ACPD.

(T26) P25, L23: Title of Stephens et al., 2002 should not be spelled in all capital letters.

Please also note the supplement to this comment:
https://www.atmos-chem-phys-discuss.net/acp-2018-589/acp-2018-589-RC1-supplement.pdf

Table A.

| CALIOP | | McMurdo ground-based lidar | | | | |
|---|---|---|---|---|---|---|
| | | STS | NAT mixtures | Enhanced NAT | Ice | No PSC |
| | STS | ** % | ** % | ** % | ** % | ** % |
| | NAT mixtures | ** % | ** % | ** % | ** % | ** % |
| | Enhanced NAT | ** % | ** % | ** % | ** % | ** % |
| | Ice | ** % | ** % | ** % | ** % | ** % |

Table B.

| McMurdo Ground-based lidar | | CALIOP | | | | |
|---|---|---|---|---|---|---|
| | | STS | NAT mixtures | Enhanced NAT | Ice | No PSC |
| | STS | ** % | ** % | ** % | ** % | ** % |
| | NAT mixtures | ** % | ** % | ** % | ** % | ** % |
| | Enhanced NAT | ** % | ** % | ** % | ** % | ** % |
| | Ice | ** % | ** % | ** % | ** % | ** % |

Table A shows the statistics when CALIOP measured specific class of PSC, what PSC was observed by McMurdo ground-based lidar, or no PSC was observed.   Table B shows the statistics when ground-based lidar measured specific class of PSC, what PSC was observed by CALIOP, or no PSC was observed.

**Fig. 1.**

---

## Referee Comment (RC3) · Anonymous Referee #2 · 2 Oct 2018

**Referee Report on "Comparison of Antarctic polar stratospheric clouds observations by ground- and satellite based lidars and relevance for Chemistry Climate Models" by Snels et al.**

**General comments**

This paper presents a statistical comparison of Polar Stratospheric Clouds (PSCs) occurrences for different Antarctic PSC fractions including NAT mixtures, STS, ice, and enhanced NAT mixtures between ground-based measurements and CALIPSO data. In a second step, the CALIPSO data are compared with 5 different Chemistry-Climate Models (CCMs) using several diagnostic methods, based respectively on the vertical extend of PSCs for all PSC fractions, the total PSC frequency for NAT mixtures and ice, the histogram of SAD values for all PSC fractions, and the evolution of the NAT and ice fractions as a function the difference between the temperature and the NAT equilibrium temperature.

I am not convinced that the way the authors process the CALIPSO and ground-based lidar data is always rigourous and adequate, and this might be a source of many biases and difficulties.

Further, the way to evaluate the agreement between the CALIOP and ground-based datasets, but also the agreement between the different models and CALIOP, look subjective in some cases (e.g. comparison CALIOP-ground-based lidar based on Figure 1, distinction between "rather good agreement above 15 km" and "biased below 15 km" on Figure 2, general rejection of "outlier" LMDZrepro model although this model scores not so bad following some specific criteria).

Concerning the comparison between CCM's and CALIPSO, I find striking that the "best model" giving the best agreement with CALIPSO is highly depending on the methodology used: Based on total PSC frequencies (Table 2), LMDZrepro and WACCM-ccmi are performing the best; based on the SAD histogram, LMDZrepro shows the best agreement based on the range of $Log_{10}(SAD)$; WACCM and CAM3.5 give the closest evolution of the NAT and ice fraction as a function of $T-T_{NAT}$. Hence, CCSRNIES is the only one of the 5 models considered here that cannot pretend to the status of "best model" following any diagnostic method, although the authors reject overall another model, namely LMDZrepro, and outlier. Overall, I don't see any clear conclusion from this work, and my general feeling is mainly that the way the CALIPSO data ground-based lidar data are processed might present biases or be inadequate, and that the implementation of the different diagnostic methods should be improved.

Finally, the text is often lacking in rigour or written in a language punctuated by approximate expressions and mistakes, making the reading sometimes very difficult. This should be improved.

**Detailed comments**

**Abstract**

- L. 3-5, p.1: This sentence is particularly difficult to read. Please reword in a more fluent way.
- L. 1 and 6, p.1: The authors repeat partly the same idea. The text could be written more efficiently, or in another way to put the emphasis on the main focus of the sentence.

**1. Introduction**

- L. 7-8, p.2: "Many different schemes…": Do the authors mean that the different schemes use different thresholds for detection and classification ?
- L.11-12, p.2: "Ground-based lidar observatories… from the early nineties to today": The authors might be only interested by the period from the early nineties until today, or by a specific location (probably McMurdo), but there exist ground-based lidar time series spanning at least 2 decades more ! (See for instance Jäger, J. Geophys.Res., 2005). Hence, they should be more specific.
- L.12-13, P.2:" A clear issue …": Do the authors mean that the ground-based time series above Antarctica are not representative enough for climatological studies and model evaluation above Antarctica ? This should require a reference.

**2. Comparison of PSC observations by ground-based and satellite based lidars**

**2.1 CALIPSO observations**

**2.2 Ground_based PSC observations at McMurdo**

- L.20, p.3: "Klett algorithm": This requires a reference.
- L.2-3, p.4: What do the authors mean by "facilitate" ? Is it about reducing the dataset ? Or having a regular time base ? Or something else ?

**2.3 PSC detection and classification**

- L. 24, p.4-l. 8, p.5: The authors are restarting an overview of the literature, citing the same works as in the overview literature in the introduction. This cares for unnecessary repetitions. The authors should focus on the message needed at this point of the discussion, without repeating what was said before.
- L.1-2, p.5: These lines include 2 almost similar sentences about the same work ! Please remove what is not necessary.
- L. 1-6, p.5: The same reference is cited 3 times during the description of this work. Please remove two of them !

**2.4 PSC detection and classification criteria for the CALIPSO V2.0 data**

- L. 10-12, p.5: Here again, the authors repeat what has been written in the introduction (on ll. 8-10, p.2).
- L.13, p.5: "below" is actually immediately after the sentence. "As follows" might be more appropriate.
- L. 14, 16, p.5: The use of "now" brings some confusion: do the authors mean "in Version 2" or "in the present work" ? Using "In Version 2" (if this is what is meant) might clarify this point.
- L.17-19, p.5: These two sentences are difficult to read. Do the authors mean that there are two criteria, and that a PSC occurrence is assumed if at least one of the criteria are fulfilled ? Writing that two threshold for background aerosols, respectively for the perpendicular backscatter and the scattering ratio, are defined as their median value plus one median deviation, might already clarify the text. Using formulas might also make it more clear.  It is also not clear for me what is the relationship between the median deviation and the "unc" quantity. I understand from the text that, in both cases, the effective threshold is the median value+median deviation+ uncertainty. Is it what the authors mean ? Again, an expression using an equation may remove any ambiguity.
- L.2, p4; l.17, p.5; l.30, p.6: the time references are confusing. In l.2, p.4, it is indicated that about 1 data point estimated from 30 minute observation is considered every 6h at most; In l.30, P.6, this becomes "1 or 2 measurements occurring per day". And in l. 17, p.5, the authors consider a "daily median". On which sampling do they compute the median ? And does the explanation in p.5 mean that a different threshold is considered every day ? An hence that the "background value" is changing every day ? This seems a strange concept of "background value" !
- L. 20-31, p.5: Again, all this long description of PSC types would be much more easy to read if they were included in a table and supported by some equations in the text.  Also, if the authors find necessary to repeat the change of criteria performed in the CALIPSO dataset, they should at least explain why all these changes are made. Is it a response to the conclusions of the work by (Pitts et al., 2018) explained in ll. 3-6, p.5 ? If yes, the conclusions of (Pitts et al., 2018) might be moved to here.
- L. 26-29, p.5: I understand that MLS is used to select the PSC type observed by CALIPSO, and that CALIPSO is used to determine the selection criteria. Is there here any problem of snake biting its own tail ? How effective is then this selection ?
- L. 32, p.5: "the PSC classified grid": What does it mean ?
- L. 32, p5: Which optical parameters ?

**2.5 PSC detection and classification criteria for the ground-based data**

- L. 5-9, p.6: Here, the threshold for PSC detection are clearly constant. In which extend are these criteria consistent with the criteria used in ll. 17-19, p.5 ?

- L. 11-13, p.6: I am not sure if this selection occurs in the same way as for the CALIPSO data (See L. 25-26, p.5). Which is the criteria used in that case and how consistent are the selection criteria for the CALIPSO data and the ground-based data ?
- L.13, p.6: Why do the authors consider here monthly averages while they consider daily averages before ? Isn't there a lack of coherence in their choices?
- L. 4-15, p.6: Again, using a table for all the selection criteria could be more readable and make the comparison with equivalent selection criteria applied to CALIPSO more readable.

**2.6 Comparison of coincident PSC observations at McMurdo from the ground and from CALIPSO during the 5-year observation period**

- L. 19, p.6: What do the authors mean by "unique definitions" ? Here, the criteria used for ground-based and CALIPSO measurements are different !? This sentence sounds also not very fluent.
- L. 3-4, p.7: Does it means that the criteria provided in §2.4, specifically for CALIPSO, are actually not the ones that are really used ? This is quite confusing !
- L. 8, p.7 – l.11, p.9 and Table 1: It is extremely difficult to conclude that the agreement between both plots is good. When focusing on very limited periods showing a clear pattern related to a specific PSC type on one of the plots, the other plot often doesn't show a similar pattern at the same time and same altitude range. Hence, I cannot agree with the statement in l.6, p.8, that "the overall agreement is rather good".  The authors try to confirm the agreement by providing a statistical comparison over 5 year: this is quite a long time, and I don't think that the relatively good agreement found between ground-based and CALIPSO for STS, NAT mixtures and ice may provide any real evidence of the agreement between both datasets. I guess it rather gives an overall probability to find a specific PSC type above Mc Murdo, which is something quite different. For the enhanced NAT mixtures, the situation is even worse since there is about a factor of 2 between the statistics, despite the long time period. Results presented in Figures 2 and 3 are also calculated as averages over a five-year time period, so that they don't bring more evidence on the agreement between ground-based and CALIOP measurements. Hence, as suggested by the authors higher in the text, the difference in measurement rate and coverage, different geometry and measurement protocols may induce significant biases in the PSC classification. Did the authors compare directly coincident measurements at specific very limited periods ? Even if, as explained by the authors in l.5-6, p.7, a point-to-point profile comparison may be unsatisfactory, we should expect that a comparison within a short period shows similar patterns in both plots.
- L. 3, p.8: "at the core of the PSC winter season": it might be useful to mention the corresponding period in terms of months.

- L. 1-5, p.11: I don't see how the different geometries could justify the differences in the results, since Figure 2 presents PSC fractions, and not absolute values. It can be argued that CALIPSO will be more sensitive at high altitude and the ground-based lidars at lower altitude, but I guess this applies to all kinds of PSC. Hence, it is conceivable that the total number of observed events could be affected, but probably not the PSC fractions. Concerning the differences in statistics, how do the authors expect them to influence the agreement between datasets ?
- L. 3-4, p.12, Figures 2 and 3: What can explain that the the temperature dependence of the NAT fraction max agree quite well between CALIPSO and ground-based measurements (Figure 3), while the same NAT fraction are so different at some altitudes, e.g. around 20-22 km (Figure 2) ? It is unlikely that the number of events is too small at these altitudes to make the estimated fractions statistically not significant.
- L. 8-10, p. 12: I don't understand this conclusion: the differences are manifest on Figure 2.

**3. Comparison of CALIOP PSC observations in the Southern Hemisphere with CCM simulations**

- L. 17-31, p.12: The resolution should be mentioned for the different models and datasets. Resolution aspects play most probably a crucial role in the comparison between models, and with CALIPSO (See also comments on L.4, p.17 and Figure 7).
- L.14-15, p.13: Which kind of threshold do the authors apply to the SAD when applying the observation operator ? Do the authors mean that they use a mask recording the amount of lidar measurements in every grid cell and putting to zero all grid points that are not covered by any lidar presence ?
- L. 16, p.13: The formulation is confusing: is "the sum of all layers" an amount of layers or a distance in km (= amount of layers x 1.5 km) ?
- Caption Figure 4: "the number of km": Please be more specific: does it concern the altitude range ?
- L. 6, p.14: What do the authors mean by "NAT-like" ? The ensemble NAT mixtures + enhanced NAT mixture ?
- L.1, p.17: Are there no reasons to think that it is the CALIPSO PSC frequencies that are underestimated with respect to the reality ? I have in mind the way the statistics are processed, the use of monthly means, and the characteristics of the CALIPSO/ground-base station coverage.
- L.4, p.17 and Figure 7: "a very large underestimation": with respect to what ? In July, it is very similar to WACCM-cmmi, and very similar to WACCM in August. In September, LMDZrepro is much larger than WACCM. The "very large underestimation" is certainly not general when considering the total PSC frequency. However, it is true when considering the SAD criteria (Figure 7). It has to be noted that LMDZrepro gives overall the closest to CALIPSO in both cases (Total PSC frequency and SAD). Would the similarity with

CALIPSO and the outlier character with respect to the other models in the case of the SAD diagnostic be related to the coarser grid resolution of the LMDZrepro model with respect to the other models ?

- L.5, p.17: "The largest biases are found for ice PSCs that tend to be significantly overestimated": Do the authors mean: "underestimated" ? I guess they are still considering the LMDZ model ?
- L. 7-8, p.17: Taking into account the difference in assumptions, what is the reliability and the robustness of such diagnostic method ? A sensivity study might be needed.
- L. 6, p.18: "This in turn would give less irreversible denitrification processes than in the case of simulation by the models with larger NAT SAD" ?
- L.4, p.19: occurences of what ? Please be more specific.
- L. 6, p. 19: How is the averaging performed ? As a simple mean of all numbers ? Or by weighting by the grid cell area ? Concerning CALIPSO, how do the authors use the monthly means ? By making a mean of means ? Averaging yet averaged values may affect significantly the results.
- L. 10-12, p.19: "Too slow", "too fast": with respect to CALIPSO ? This should be specified. What do the authors mean by "progression for ice/NAT" ?
- L. 1, p.20: "The fraction of data with different PSC": Please revise the formulation.
- L. 3, p.20: the fraction of what ? Please be specific ! "an increase of ice with T-TNAT < -5K": Please revise the formulation: increase with decreasing temperature.
- L.5, p.20: "a sharper increase of the fraction": fraction of what ?
- L. 7, p.20: "while for the other models, the ice…".

**4. Conclusions**

- L. 12, p.20: A point-to-point comparison is always feasible ! The issue is to know if it is valid and reliable.
- L. 14, p. 20: "very similar": Based of the results presented in Figure 1, I don't agree. (See comment above). At least, a statistical indicator and quantitative estimates of the uncertainty should provided.
- L. 16, p.20: As already mentioned, I don't understand the emphasis on "below15 km". Is it based on Figure 2; If well, this seems very subjective to me.
- L. 16-17, p.20: "rather good above 15 km": this looks particularly subjective. At least, the "rather good" should be quantified in some way !
- L. 20, p.20: "Models fail to reproduce realistic geographical distributions of PSCs": I am really not convinced by the demonstration made in this paper. A significant part of the problem might come from the way the authors implement their different methodologies, and more particularly from the comparison of things that are not really comparable.
- L. 22, p.20: The more recent WACCMI-ccmi model compared better with CALIOP only for one specific diagnostic method (based on the total PSC

frequency). The issues is to understand why: in view of all my previous criticisms, it might be fortuitous.

**Technical corrections**

- L. 32, p.2: "The most recent…": "CCM" could be introduced at the first occurrence of the expression "Chemistry Climate Models", in L. 15.
- L.11, p.3: the acronym CALIOP has already been expanded above.
- L.2, p.4: "acquisition".
- L.18, p.4: remove one "of".
- L. 14-20, p.5: Putting "Data pre-processing", "PSC detection", and "PSC composition" in subtitles, or putting the ensemble in a table might simplify the layout and make the reading easier.
- L. 32, p.5: "The dataset provides" ?
- L.4, p. 6: I am not sure that "reelaborated" is correct English.
- L. 9, p.6: "ground-based".
- L. 13, p.6: "corresponding".
- L. 16, p.6: "5-year".
- L. 18, p.6: "the differing measurement procedures used for each of them" might be more correct.
- L.19, p.6: [the procedures] "induce".
- L. 2 and 6, p.7: "signal-to-noise ratio".
- L. 8, p.11: I guess the sentence is incorrect. What is "the it" ?
- L.17, P.11: I am not sure that $T_{NAT}$ is defined yet.
- L.1, p.12: "a similar behaviour".
- L.6, p.13: I guess the latitude range mentioned concern the southern hemisphere. Latitude values have thus to be indicated with a "S" or with negative numbers.
- Caption Figure 4: Missing point at the end of the sentence.
- L. 1, p.14: "90°-0°":Please specify: W or E ?
- L. 6, p.14: "O°90°": Please adapt the notation (cf. remark on L. 1, p.14 + use "-").
- L. 1, p14-l.7, p.15: It could ease the reading to use separated paragraphs for each PSC type.

---

## Author Comment (AC1) · 27 Nov 2018

The answers have been uploaded as a pdf file. See below. Answers to referee 1
A Review of "Comparison of Antarctic polar stratospheric cloud observations by ground-based and spaceborne lidars and relevance for Chemistry Climate Models" by M. Snels et al.

<General Comments>

This paper describes the comparison between PSC measurements at Antarctic Mc-Murdo Station from ground based lidar and CALIOP satellite measurements. Furthermore, the paper tries to extend the comparison of PSC statistics from CALIOP with several CCM model results from CCMVal-2 and CCMI. Although scientific value of this study might be significant, the method of comparison especially with CCM models is not well organized to derive scientifically useful conclusions, as is pointed out below. Also, there are too many typos and careless mistakes in the draft. A major revision is required before this paper will be published in ACP. I recommend that authors should check the draft carefully, including the native check, before submitting the revised draft.

(M1) In Section 3.2, the authors try to compare the PSC statistics from 5 years (2006-2010) measurements by CALIOP, with the result of 4 CCM models from CCMVal-2, and one CCM model from CCMI. However, the model run type they chose for CCMVal-2 models are REF-B2, which are targeted to be used for future predictions until 2100. The major problem for this comparison is that the result of REF-B2 run contains both inaccuracy in modeled temperatures and imperfectess in PSC schemes which are different in each model. The combination of inaccuracies both in modeled temperature and PSC schemes makes it extremely difficult to understand the nature of PSC in each model. Rather than comparison with CCMVal-2 REF-B2 runs, it is strongly preferred to compare with CCMI outputs with refC1SD runs (which is available from http://badc.nerc.ac.uk/browse/badc/wcrp-ccmi/data/CCMI-1/output), which use nudging with more realistic temperature and wind field, just to test the PSC scheme in each model. Even if the authors stick to the comparison with CCMVal-2 model results, they

should at least use the REF-B1 model run results, which are targeted to reproduce the past. In this case, the comparison with CALIOP could be made only for 2006, because REF-B1 run was made only for 1960-2006. Since CCMI refC1SD runs cover until 2010, I strongly recommend making comparisons with CCMI model outputs with CALIPSO Measurements. ANSWER: (M1) The indication of the REF-B2 run was a typing error, we apologize for that. In this manuscript we evaluate the REF-B1 simulations available for the period 1960–2006. As the reviewer highlights, those simulations were chosen because they have been constructed to include the interannual variabilities of the 11 year solar cycle, the QBO, Sea Surface Temperature (SST), volcanic effects, greenhouse gas (GHG) concentrations, and ozone‐depleting substance (ODS) concentrations (Morgenstern et al., 2010). The SST and sea ice evolutions are prescribed using the HadISST1 (Rayner et al., 2003). The variations of the GHGs and the ODSs follow the IPCC SRES A1B scenario and WMO‐adjusted scenario A1. To our opinion these free running simulations are the most suitable to be compared with the statistics from available observations.

(M2) In Section 3.1, the authors mention about more sophisticated.0/SD/CARMA model and EMAC/MSBM model, which use more realistic parameterizations for PSCs. It would gain the value of this paper significantly if they could include the comparison of CALIOP PSC statistics with the result of these models. ANSWER: The more sophisticated models are mentioned in the manuscript because those are, to our knowledge, the most significant advancements in the field of PSC representation in Global Climate Models used for the ozone and climate change studies. The CARMA model is an interactive aerosol and radiation model fully coupled to the WACCM, able to fully simulate advection, diffusion, sedimentation, deposition, coagulation, nucleation and condensational growth of atmospheric aerosols online with the temperature, dynamics and radiation structure simulated by the GCM. This approach is completely different from the parametrizations available in the simulations we are analysing here. A full evaluation of the WACCM/CARMA models in Specified Dynamics runs w.r.t. CALYPSO data are available in literature (in the Zhu et al cited works) and are outside the scope of this

intercomparison (where we work with free running simulations). It would be certainly interesting to apply the diagnostics proposed within our analysis to a free-running set of simulations performed with models including interactive aerosols. This could be the objective of a future study, when a set of simulations from new generation models might be available.

M3) In each model, denitrification and dehydration are included as is shown in Table 3. This would change the vertical distribution of $HNO_3$ and $H_2O$, which would affect the threshold temperature of NAT and ice PSCs, i.e., T_NAT and T_ice. However, this effect is never mentioned or discussed in the manuscript. Moreover, in many places in the text (especially in Sections 2.6 and 3.4), it is not clearly stated which temperature (MERRA-2, NCEP, or derived T in CCM) is used, and how T_NAT and T_ice are calculated (using $HNO_3$ and $H_2O$ value from MLS data, modeled value in CCM, or fixed values like 6 ppbv $HNO_3$ and 4.5 ppmv $H_2O$). The effect of denitrification/dehydration in modelled PSC should be discussed in the manuscript. ANSWER: First of all, we use MLS values for HNO3 and H2O concentrations, to calculate the formation temperature of NAT and ice. The temperatures used in this work are taken from MERRA-2. The temperatures used in the CCM models are generated by the models themselves, Tnat AND Tice have been calculated from the HNO3 and H2O taken from GOZCARDS

(M4) For a PSC classes comparison described in Table 1, although the percentage of each PSC class is similar, this does not prove that each one to one PSC is simultaneously observed both by ground-based lidar and by CALIOP. I would recommend authors to add the statistics showing one to one correspondence of comparison of PSC classes observed by tables like the attached tables. Table A shows the statistics when CALIOP measured specific class of PSC, what PSC was observed by McMurdo ground-based lidar, or no PSC was observed. Table B shows the statistics when ground-based lidar measured specific class of PSC, what PSC was observed by CALIOP, or no PSC was observed. ANSWER: It is not the goal of the article to make

a point-to-point comparison for validation purposes. The goal is to verify if the ground-based measurement are representative for a larger area, typically contained in a 7x2 degrees box around McMurdo. Apart from that a point-to-point analysis presents the following difficulties: 1) None of the overpasses of CALIPSO are sampling the same air mass as the ground based lidar. To illustrate this I show a plot of all overpasses within the 7x2 degrees box, which corresponds roughly to a distance of 100 km from . McMurdo. While CALIOP provides a resolution of 5 km ( when integration is required due to low signal-to-noise ratio up to 135 km !) the air mass sampled by the ground-based lidar extends to at most 100 m. (30 km * 3 mrad field of view of the telescope). Another important difference of the two lidars is that a CALIOP overpass occurs in about 30 seconds, while the ground-based data are integrated over 30 minutes. This implies that the ground-based measurement integrates air masses moving with a wind speed varying from 0 to 50 m/s, depending also on the altitude (the wind speed might be very different at 15, 20 and 25 km), rendering a comparison with an instantaneous profile of CALIOP very questionable. However, the statistical analysis is only meaningful if the sampling of the two lidars covers the same period of time and if this period of time has a dense coverage. In order to achieve this we concentrate on 2006, having a large number of observations by both lidars with a good coverage (see figure 1 of the manuscript). We then analyse the months July and August and report the statistics in terms of occurrences of PSC classes and dependence on altitude.

Differences and agreement have been discussed in the revised manuscript.

(M5) In Section 3.4, they discuss about the cold pole bias in most CCMVal-2 CCM models. However, when I read the SPARC report No.5 Chapter 4 "Section 4.3.5 Polar stratospheric cloud threshold temperatures" in page 128, there is an explanation that CCM models have warm bias and A_NAT and A_ice show low value compared with ERA-40 temperature. This description totally contradicts with the discussion described in Section 3.4. Please explain why such contradiction occurs.

ANSWER: Looking at figure 4.1 of the Sparc report (page 112) it is evident that all models have a cold temperature bias except for the two UMUKCA models. This is explicitly stated on page 113. Figure 4.15 in "Section 4.3.5 Polar stratospheric cloud threshold temperatures" in page 128, shows that the same two models strongly underestimate the mean PSC Area's which is of course in agreement with the warm bias of these models discussed before. So there is no contradiction.

All the corrections suggested by the referee below have been made.

ïČij (S1) The numbers in author list are not ordered correctly, i.e., 1, 5, 2, 3, 4. It should be

ïČij 1, 2, 3, 4, 5.

ïČij (S2) P1, L3: The abbreviation of CALIOP should be shown also in the abstract.

ïČij (S3) P1, L9: The meaning of "... and a selection simulations obtained ..." is unclear.

ïČij (S4) P1, L4: In Pitts et al. (2018, ACP), they use "v2" instead of "V2". Please check if

ïČij V2 should be changed to v2 throughout the manuscript or not.

ïČij (S5) P1, L18: The abbreviation of WACCM-CCMI should be shown.

ïČij (S6) P2, L7: The abbreviation of CALIOP should be shown here, not at P2, L20.

ïČij (S7) P2, L18: Chemistry Climate Models –> Chemistry Climate Models (CCMs)

ïČij (S8) P2, L20: clouds and aerosol –> clouds and aerosols

ïČij (S9) P2, L26: Chemistry Climate Models –> CCMs

ïČij (S10) P2, L29: The SPARC Report No5 (2010) cannot be found in the reference list.

ïČij (S11) P2, L30: Chemistry Climate Models –> CCMs

ïČij (S12) P3, L1: Chemistry Climate Models (CCM) –> CCMs

ïČij (S13) P3, L14: CALIOP (Cloud Aerosol Lidar with Orthogonal Polarization) –> CALIOP

ïČij (S14) P3, L14: Details on CALIOP –> Details of CALIOP

ïČij (S15) P4, L16: Reference (Cairo et al., 1999) should appear at the end of Line 18.

ïČij (S16) P5, L14: CALIPSO V2.0 data –> CALIPSO v2 data

ïČij (S17) P5, L15: V2.0 –> v2

ïČij (S18) P5, L17: V1.0 and V2.0 –> v1 and v2

Please also note the supplement to this comment:
https://www.atmos-chem-phys-discuss.net/acp-2018-589/acp-2018-589-AC1-supplement.pdf

———————————————

[Figure]

Fig. 1.

[Figure]

[Figure]

**Fig. 2.**

[Figure]

**Fig. 3.**

---

## Author Comment (AC2) · 27 Nov 2018

Answers to Referee 2

First of all, we want to remark that the referee is referring to a different version of the paper with respect to the one posted on the web-site; acp-2018-589.pdf, probably to the version submitted on 12/06/2018, prior to publication on the website. We thank the referee for his very constructive review, which has surely improved the paper. General comments I am not convinced that the way the authors process the

[Figure]

CALIPSO and ground-based lidar data is always rigourous and adequate, and this might be a source of many biases and difficulties. Further, the way to evaluate the agreement between the CALIOP and ground-based datasets, but also the agreement between the different models and CALIOP, look subjective in some cases (e.g. comparison CALIOP-ground-based lidar based on Figure 1, distinction between "rather good agreement above 15 km" and "biased below 15 km" on Figure 2, . . ...

ANSWER: To answer the referee's lack of confidence in the correctness of the lidar data processing and in order to convince him of the correct treatment of the data, we first state that the CALIOP data have been used as provided by the PI's, using the v2 version of the classified PSCs. The detection and the classification of the ground-based data has been explained in more detail and a new figure has been added to illustrate how the detection and classification algorithm works.

While the value of the confidence indexes provides the confidence in the classification, its value is not used in the classification algorithm, and it provides only a threshold value between two classes. Therefore we've eliminated the confidence indexes from the manuscript and discuss the classification algorithm in terms of threshold values (see figure above). These threshold values have been determined in some cases differently for the two lidars, due to the different nature of the data they produce. This has been discussed in the revised manuscript. For instance, the threshold values for R and $\delta_{perp}$ correspond with background aerosols, observed in absence of PSCs. These can be easily determined from the CALIOP data, producing daily values, by considering PSC area's on the southern hemisphere at temperatures above 200 K. For the ground-based lidar it is not possible to obtain daily values, and an average has been made of PSC free observations in early June and October.

Several bugs have been found in the normalization of the data, thus producing wrong values for the fraction of the PSC classes. The figure shown above shows the new values for 2006. As a consequence the discussion has been adapted and the distinction between below and above 15 km has been eliminated. A part of the revised
manuscript: The figure shows that PSCs are observed up to 25 km in July and August. Above 25 km the number of PSC observations is negligible, both for ground-based and CALIOP observations. NAT mixtures are the dominating species in July and August, with a slightly different altitude distribution in July; ground-based occurrences of NAT mixtures are more frequent below 18 km with respect to CALIOP data. The occurrences of ice clouds in July are very similar, while in August some low ice clouds appear in the ground-based data, but are absent in the CALIOP observations. Enhanced NAT mixtures occur mainly in July, and are observed between 17 and 25 km, more abundant in the ground-based observations. The vertical distribution of STS shows a good agreement in July and August.

. . ..general rejection of "outlier" LMDZrepro model although this model scores not so bad following some specific criteria). Concerning the comparison between CCM's and CALIPSO, I find striking that the "best model" giving the best agreement with CALIPSO is highly depending on the methodology used: Based on total PSC frequencies (Table 2), LMDZrepro and WACCM‐ccmi are performing the best; based on the SAD histogram, LMDZrepro shows the best agreement based on the range of Log10(SAD); WACCM and CAM3.5 give the closest evolution of the NAT and ice fraction as a function of T‐TNAT. Hence, CCSRNIES is the only one of the 5 models considered here that cannot pretend to the status of "best model" following any diagnostic method, although the authors reject overall another model, namely LMDZrepro, and outlier. Overall, I don't see any clear conclusion from this work, and my general feeling is mainly that the way the CALIPSO data ground‐based lidar data are processed might present biases or be inadequate, and that the implementation of the different diagnostic methods should be improved. ANSWER: The reviewer is correct, the previous version of the text was giving the impression of a general scoring of the models, with a final "negative" score for the LMDzRepro or the idea to derive a "best model". This is not the scope of the manuscript. The main focus here is to define diagnostics that permits to compare observations with the "model world" in a consistent way. In order to disentangle, when possible, biases deriving from specific parameterizations that could be

attenuated in principle with future improvement, and biases related to the global biases of the model and more difficult to target. For example, when the error is strongly associated to the cold pole bias in stratospheric temperature and therefore attributed to model dynamics, it requires a more structural intervention on the model definition than when bias is associated to the assumptions in the specific parametrization made on the number of particles per cm3. A future study might imply the development of specific metrics, derived from the diagnostics proposed here, that could allow to define scores and evaluate models. However, as the reviewer correctly remarks, this would not be a straightforward way of proceeding and it is outside of the scope of the present work. We have adjusted the text in relevant sections to illustrate this.

Detailed Abstract L. 3‐5, p.1: This sentence is particularly difficult to read. Please reword in a more fluent way. ANSWER: The sentence has been divided in two pieces in order to facilitate the reader. L. 1 and 6, p.1: The authors repeat partly the same idea. The text could be written more efficiently, or in another way to put the emphasis on the main focus of the sentence. The sentence has been re-edited. Below follows the new text: Abstract. A comparison of polar stratospheric clouds (PSCs) occurrence from 2006 to 2010 is presented, as observed from the ground-based station McMurdo (Antarctica) and by the satellite-borne CALIOP lidar (Cloud-Aerosol Lidar with Orthogonal Polarization) measuring over McMurdo. McMurdo (Antarctica) is a primary station in the NDACC (Network for Detection of Atmospheric Climate Change). The ground-based observations have been classified with an algorithm derived from the recent v2 detection and classification scheme, used to classify PSCs observed by CALIOP. A statistical approach has been used to compare ground-based and satellite based observations, since point-to-point comparison is often troublesome due to the intrinsic differences in the observation geometries and the imperfect overlap of the observed areas.

1. Introduction L. 7‐8, p.2: "Many different schemes...": Do the authors mean that the different schemes use different thresholds for detection and classification ?

ANSWER: The text has been modified; indeed the different schemes often use different thresholds. Many different schemes using thresholds for detection and classification have been proposed, rendering a comparison difficult.

L.11‐12, p.2: "Ground‐based lidar observatories. . . from the early nineties to today": The authors might be only interested by the period from the early nineties until today, or by a specific location (probably McMurdo), but there exist ground‐based lidar time series spanning at least 2 decades more ! (See for instance Jäger, J. Geophys.Res., 2005). Hence, they should be more specific.

ANSWER: We refer to lidar observations in Antarctica. Anyway we now have included also the earliest, up to our knowledge, lidar observations in Antarctica, with references, from 1985 on. Of course there exist ground-based lidar observations much earlier, but not in Antarctica. The Jaeger paper deals with observations in Garmisch-Partenkirchen. The first lidar observations in Antarctica started in 1985 at Syowa Station. Iwasaka and co-workers (Iwasaka, 1985, 1986) used a polarization sensitive lidar to measure backscatter and depolarization to observe PSCs. Later, in 1987/1988 at the Amundsen-Scott South Pole Station, Fiocco and co-workers (Fiocco et al., 1992) used the elastic backscatter signal from a 20 lidar operating at 532 nm to observe PSCs in relation to the temperature. PSCs have also been observed at Davis, from 2001 to 2004 (Innis and Klekociuk, 2006) and at Rothera (Simpson et al., 2005) from 2002 to 2005. Long-term observations of PSCs have been performed at McMurdo (Adriani et al., 1992, 1995, 2004; Di Liberto et al., 2014), from 1989 until 2010 and at Dumon D'Urville (Santacesaria et al., 2001; David et al., 1998, 2010), from 1990 until now, both with polarization sensitive lidars. Recently the McMurdo lidar has been transferred to Dome C and is operating there 25 from 2014 on (Snels et al., 2018).

L.12‐13, P.2:" A clear issue . . .": Do the authors mean that the ground‐based time series above Antarctica are not representative enough for climatological studies and model evaluation above Antarctica ? This should require a reference. ANSWER: The Antarctic lidar stations are few and those with a long term record even fewer (Mc-

Murdo, Dumont D'Urville and Dome C). This means that model calculations can be compared at a few locations. It doesn't mean that they are not representative enough for climatological studies.

2. Comparison of PSC observations by ground‐based and satellite based lidars 2.1 CALIPSO observations 2.2 Ground_based PSC observations at McMurdo L.20, p.3: "Klett algorithm": This requires a reference. ANSWER: A reference for the Klett algorithm has been added. L.2‐3, p.4: What do the authors mean by "facilitate" ? Is it about reducing the dataset ? Or having a regular time base ? Or something else ? ANSWER: It means that we would like to compare data on a daily base, since CALIOP produces at most one overpass per day. Thus we proceed as follows: if more than one ground-based profile is available within a 6 hour time window, only the profile with the smallest time difference with respect to the Calipso overpass is considered. However, this situation is rarely verified. We explained better in the text how we obtain a daily profile for the ground-based data.

2.3 PSC detection and classification L. 24, p.4‐l. 8, p.5: The authors are restarting an overview of the literature, citing the same works as in the overview literature in the introduction. This cares for unnecessary repetitions. The authors should focus on the message needed at this point of the discussion, without repeating what was said before. ANSWER: The title of this paragraph justifies a reference to the recent review by Achtert and Tesche. in our opinion. The detection scheme used in this work is based on the CALIOP algorithms, so it is obvious that these are mentioned here.

L.1‐2, p.5: These lines include 2 almost similar sentences about the same work ! Please remove what is not necessary. ANSWER: The sentence has been removed L. 1‐6, p.5: The same reference is cited 3 times during the description of this work. Please remove two of them ! ANSWER: The three references have been removed and we now refer only to Pitts2018, for the V2 classification.

2.4 PSC detection and classification criteria for the CALIPSO V2.0 data L. 10‐12,

p.5: Here again, the authors repeat what has been written in the introduction (on ll. 8‐10, p.2). ANSWER: The sentence has been removed and the text has been modified.

L.13, p.5: "below" is actually immediately after the sentence. "As follows" might be more appropriate. ANSWER: "Below" has been substituted with "as follows" as suggested by the referee

L. 14, 16, p.5: The use of "now" brings some confusion: do the authors mean "in Version 2" or "in the present work" ? Using "In Version 2" (if this is what is meant) might clarify this point. ANSWER: "now" has been substituted with "in Version 2" as suggested by the referee

L.17‐19, p.5: These two sentences are difficult to read. Do the authors mean that there are two criteria, and that a PSC occurrence is assumed if at least one of the criteria are fulfilled ? Writing that two threshold for background aerosols, respectively for the perpendicular backscatter and the scattering ratio, are defined as their median value plus one median deviation, might already clarify the text. Using formulas might also make it more clear. It is also not clear for me what is the relationship between the median deviation and the "unc" quantity. I understand from the text that, in both cases, the effective threshold is the median value+median deviation+ uncertainty. Is it what the authors mean ? Again, an expression using an equation may remove any ambiguity.

ANSWER: Yes, "or" means that it is sufficient if one of the two criteria is fulfilled We rewrote this section and added a figure to better explain the detection and selection criteria.

Figure 1. The figure shows the detection and classification criteria of the V2 CALIOP algorithm. The classification as STS, NAT mixtures, enhanced NAT mixtures and ice, requires that threshold conditions for R and/or bperp are satisfied. See the text for details. The following paragraphs substitute the old ones in the manuscript: 2.4 PSC

Detection and classification criteria for the CALIPSO v2 data The CALIOP v2 PSC detection and composition classification algorithm (Pitts et al., 2018) has been used to create the recently released CALIOP v2 PSC mask database covering the period from June 2006 to October 2017. Here we compare these v2 data with ground-based observations at McMurdo from 2006 to 2010. Major enhancements in the v2 algorithm over earlier versions include daily adjustment of composition boundaries to account for effects of denitrification and dehydration, and estimates of the random uncertainties u(bperp ) and u(R) due to shot noise in each data sample, which are used to establish dynamic detection thresholds and composition boundaries. The CALIOP v2 algorithm is represented pictorially in Figure 1 and is described in more detail in the following sections.

2.4.1 PSC detection PSCs are detected in the CALIOP data as statistical outliers relative to the background stratospheric aerosol population. The v2 background aerosol thresholds bperp_;thresh and Rthresh are calculated as the daily median plus one median deviation of CALIOP data at ambient temperatures above 200 K. PSCs are those data points for which either bperp > bperp;thresh+u(b_perp) or R> Rthresh+u(R). If perp <= perp_thresh +u(perp) and R < Rthresh +u(R), the point is a non-PSC. Noise spikes are eliminated in the CALIOP v2 data by requiring coherence within a running 3-point vertical by 5-point horizontal along-track box. 2.4.2 PSC composition The PSC composition is determined as follows: • If perp <= perp_thresh +u(perp ), but R > Rthresh +u(R), the PSC is classified as STS.

• A PSC with perp > perp_thresh +u(perp ) is assumed to contain non-spherical particles and is classified as NAT (or enhanced NAT) mixture or ice based its value of R. The boundary value separating ice from NAT and enhanced NAT mixtures, RNATjice, is calculated based on the total abundances of HNO3 and H2O vapors as determined on a daily basis as a function of altitude and equivalent latitude from nearly coincident cloud-free Aura MLS data

• If perp > perp_thresh +u(perp ) and R > RNATjice, the PSC is classified

as ice.

• If 2 < R < RNATjice and ïĄ́cperp > 2_10-5m-1sr-1, the PSC is classified an enhanced NAT mixture. All other PSCs with ïĄ́cperp > ïĄ́cperp_thresh +u(ïĄ́cperp )and R < RNATjice are classified as NAT mixtures.

The CALIOP v2 data set provides both the grid of classified PSCs according to the v2 algorithm and the associated optical parameters.

L.2, p4; l.17, p.5; l.30, p.6: the time references are confusing. In l.2, p.4, it is indicated that about 1 data point estimated from 30 minute observation is considered every 6h at most; In l.30, P.6, this becomes "1 or 2 measurements occurring per day". AN-SWER: CALIOP overpasses do not occur every day and at most twice per day. In average we have about 30 CALIOP overpasses per month. Ground-based lidar data are mostly recorded during a CALIOP overpass, but also on days without CALIOP overpasses, usually at the same time that CALIOP overpasses occur and sometimes at different times from the CALIOP overpasses. The latter are not included in this analysis. All other ground-based measurements have been used in the statistical comparison. Generally speaking most of the ground-based profiles have been recorded during a CALIOP overpass, but there might be days with either a ground-based measurement or a CALIOP measurement. So we include all CALIOP measurements falling in a spatial box around McMurdo, and all ground-based data measured in a time frame dictated by CALIOP overpasses, including also the days without overpass. The text has been adapted accordingly in the revised manuscript.

And in l. 17, p.5, the authors consider a "daily median". On which sampling do they compute the median ? And does the explanation in p.5 mean that a different threshold is considered every day ? An hence that the "background value" is changing every day ? This seems a strange concept of "background value" !

ANSWER: These considerations concern the criteria for the CALIOP data. As said before the CALIOP data were used as supplied by the PIs. The criteria applied by

the CALIOP team use a median value of observations above 200 K, i.e in absence of PSCs. The background values are defined as the values of R and ïĄ́ćperp in absences of PSCs. Indeed these values can change during the season.

L. 20‐31, p.5: Again, all this long description of PSC types would be much more easy to read if they were included in a table and supported by some equations in the text. Also, if the authors find necessary to repeat the change of criteria performed in the CALIPSO dataset, they should at least explain why all these changes are made. Is it a response to the conclusions of the work by (Pitts et al., 2018) explained in ll. 3‐6, p.5 ? If yes, the conclusions of (Pitts et al., 2018) might be moved to here.

ANSWER: We inserted a figure showing in a simple way how the detections and classification algorithm uses threshold values . See also the answer given above to the general comments. The v2 algorithm has also been explained better in the text.

L. 26‐29, p.5: I understand that MLS is used to select the PSC type observed by CALIPSO, and that CALIPSO is used to determine the selection criteria. Is there here any problem of snake biting its own tail ? How effective is then this selection ? ANSWER: Cloud free means that CALIOP did not observe clouds, including PSC clouds of course. All cloud-free MLS data for HNO3 and H2O concentrations have been used to determine one of the selection (not detection !!!) criteria of Caliop

L. 32, p.5: "the PSC classified grid": What does it mean ? ANSWER: This is really confusing, we substituted with "the grid of classified PSCs"

L. 32, p5: Which optical parameters ? ANSWER: The optical parameters are; backscatter ratio, perpendicular and parallel backscatter coefficient

2.5 PSC detection and classification criteria for the ground‐based data L. 5‐9, p.6: Here, the threshold for PSC detection are clearly constant. In which extend are these criteria consistent with the criteria used in ll. 17‐19, p.5 ?

ANSWER: The huge number of data acquired by Caliop allow for a very sophisticated

statistical elaboration, including the determination of daily means for the threshold. The lidar data are in comparison very few and thus it is very difficult to obtain a reliable daily values. Therefor an average value for the threshold has been adopted, based on previous experiences and also very similar to the average threshold used in the analysis of the Caliop data.

L. 11‐13, p.6: I am not sure if this selection occurs in the same way as for the CALIPSO data (See L. 25‐26, p.5). Which is the criteria used in that case and how consistent are the selection criteria for the CALIPSO data and the ground‐based data ? ANSWER: The referee probably refers to the phrase "The discrimination between NAT mixtures and enhanced NAT mixtures is made by using the condition R > 2 and bperp > 2_10-5 m-1sr-1, while the RNAT|ice threshold has been taken from the corresponding CALIOP data, by extrapolating daily values in case of no overpass. The first part is done in exactly the same way for Caliop and ground-based data. The threshold R(NAT|ice) has been taken from the corresponding CALIOP data, by extrapolating daily values, because it is not always possible to associate a ground-based observation with a coincident Caliop observation.

L.13, p.6: Why do the authors consider here monthly averages while they consider daily averages before ? Isn't there a lack of coherence in their choices? ANSWER: This is an error. We extrapolate RNAT|ice from the CALIOP data because Caliop overpasses do not occur on every day within a distance of 100 km from McMurdo. Moreover we are comparing ground based and satellite measurements that are often, but not always, coincident in time. L. 4‐15, p.6: Again, using a table for all the selection criteria could be more readable and make the comparison with equivalent selection criteria applied to CALIPSO more readable.

ANSWER: We inserted a figure for detection and selection criteria

2.6 Comparison of coincident PSC observations at McMurdo from the ground and from CALIPSO during the 5‐year observation period

[Figure]
ANSWER: The word coincident is referring to the spatial coincidence, that is considering all measurements of both instruments falling in the box defined as ....We had eliminated the word coincident from the document, in order to avoid confusion, but apparently one escaped our attention, we apologize and substitute coincident by co-located here.

L. 19, p.6: What do the authors mean by "unique definitions"? Here, the criteria used for ground‐based and CALIPSO measurements are different !? This sentence sounds also not very fluent. ANSWER: The word "unique" has been omitted, since it is not pertinent

L. 3‐4, p.7: Does it means that the criteria provided in §2.4, specifically for CALIPSO, are actually not the ones that are really used? This is quite confusing! ANSWER: The analysis of the CALIOP data use averaging processes where the signal to noise ratio is low, and varies the threshold on both R and bperp as a function of signal-to-noise ratio. It does not mean that the criteria change, but that other criteria are applied as well, the so-called coherence criteria, taking into account all measured profiles on a piece of the orbit ( 5-15-45-135 km). It does not influence the analysis of the ground-based data of course.

L. 8, p.7 – l.11, p.9 and Table 1: It is extremely difficult to conclude that the agreement between both plots is good. When focusing on very limited periods showing a clear pattern related to a specific PSC type on one of the plots, the other plot often doesn't show a similar pattern at the same time and same altitude range. Hence, I cannot agree with the statement in l.6, p.8, that "the overall agreement is rather good". The authors try to confirm the agreement by providing a statistical comparison over 5 year: this is quite a long time, and I don't think that the relatively good agreement found between ground‐based and CALIPSO for STS, NAT mixtures and ice may provide any real evidence of the agreement between both datasets. I guess it rather gives an overall probability to find a specific PSC type above McMurdo, which is something quite different. For the enhanced NAT mixtures, the situation is even worse since there is about

a factor of 2 between the statistics, despite the long time period. Results presented in Figures 2 and 3 are also calculated as averages over a five‐year time period, so that they don't bring more evidence on the agreement between ground‐based and CALIOP measurements. Hence, as suggested by the authors higher in the text, the difference in measurement rate and coverage, different geometry and measurement protocols may induce significant biases in the PSC classification. Did the authors compare directly coincident measurements at specific very limited periods ? Even if, as explained by the authors in l.5‐6, p.7, a point‐to‐point profile comparison may be unsatisfactory, we should expect that a comparison within a short period shows similar patterns in both plots. ANSWER: It is not the goal of the article to make a point-to-point comparison for validation purposes. The goal is to verify if the ground-based measurement are representative for a larger area, typically contained in a 7x2 degrees box around McMurdo. Apart from that a point-to-point analysis presents the following difficulties: 1) None of the overpasses of CALIPSO are sampling the same air mass as the ground based lidar. To illustrate this I show a plot of all overpasses within the 7x2 degrees box, which corresponds roughly to a distance of 100 km from .

While CALIOP provides a resolution of 5 km ( when integration is required due to low signal-to-noise ratio up to 135 km !) the air mass sampled by the ground-based lidar extends to at most 100 m. (30 km * 3 mrad field of view of the telescope). Another important difference of the two lidars is that a CALIOP overpass occurs in about 30 seconds, while the ground-based data are integrated over 30 minutes. This implies that the ground-based measurement integrates air masses moving with a wind speed varying from 0 to 50 m/s, depending also on the altitude (the wind speed might be very different at 15, 20 and 25 km), rendering a comparison with an instantaneous profile of CALIOP very questionable. However, the statistical analysis is only meaningful if the sampling of the two lidars covers the same period of time and if this period of time has a dense coverage. In order to achieve this we concentrate on 2006, having a large number of observations by both lidars with a good coverage (see figure 1 of the manuscript). We then analyse the months July and August and report the statistics in

terms of occurrences of PSC classes and dependence on altitude. So we follow the suggestion of the referee and analysed short periods with a good time coverage, that is July and August 2006. The referee is correct that an overall statistics covering the 5 year period is not an indication of agreement. We stated that much in the manuscript. See also the answer above to the general comment.

L. 3, p.8: "at the core of the PSC winter season": it might be useful to mention the corresponding period in terms of months. ANSWER: We added "July and August"

L. 1‐5, p.11: I don't see how the different geometries could justify the differences in the results, since Figure 2 presents PSC fractions, and not absolute values. It can be argued that CALIPSO will be more sensitive at high altitude and the ground‐based lidars at lower altitude, but I guess this applies to all kinds of PSC. Hence, it is conceivable that the total number of observed events could be affected, but probably not the PSC fractions. Concerning the differences in statistics, how do the authors expect them to influence the agreement between datasets ? ANSWER: The different observation geometries correspond with different signal to noise ratios at different altitudes. This is valid both for the parallel and perpendicular backscatter coefficient, which constitute the detection and classification thresholds for PSCs. Obviously the PSC class with low values of perpendicular backscatter coefficient (STS) and low values for the parallel backscatter coefficients (NAT) will be more effected by the S/N ratio than ice and enhanced NAT. Since NAT and STS are the most abundant species the S/N ratio has an impact also on the PSC fractions. Moreover, it has been suggested that tropospheric meteorology and cloud cast, which hampers the ground based measurements, may also have an impact on the PSC formation above (On the linkage between tropospheric and Polar Stratospheric cloudsin the Arctic as observed by space–borne lidar, P. Achtert, M. Karlsson Andersson, F. Khosrawi, and J. Gumbel, Atmos. Chem. Phys., 12, 3791–3798, 2012www.atmos-chem-phys.net/12/3791/2012/doi:10.5194/acp-12-3791-2012)

L. 3‐4, p.12, Figures 2 and 3: What can explain that the temperature dependence of

the NAT fraction may agree quite well between CALIPSO and ground‐based measurements (Figure 3), while the same NAT fraction are so different at some altitudes, e.g. around 20‐22 km (Figure 2) ? It is unlikely that the number of events is too small at these altitudes to make the estimated fractions statistically not significant. ANSWER: We found some bugs in the program calculating the fractions. The new results have been discussed in the revised manuscript. (see also answer above to general comments).

L. 8‐10, p. 12: I don't understand this conclusion: the differences are manifest on Figure 2. ANSWER: Differences and agreement have been discussed in the revised manuscript.

3. Comparison of CALIOP PSC observations in the Southern Hemisphere with CCM simulations L. 17‐31, p.12: The resolution should be mentioned for the different models and datasets. Resolution aspects play most probably a crucial role in the comparison between models, and with CALIPSO (See also comments on L.4, p.17 and Figure 7). ANSWER: The resolution is listed in Table 2 of the published manuscript (the referee refers to another older version)

L.14‐15, p.13: Which kind of threshold do the authors apply to the SAD when applying the observation operator ? Do the authors mean that they use a mask recording the amount of lidar measurements in every grid cell and putting to zero all grid points that are not covered by any lidar presence ? ANSWER: A threshold has been defined based on the detection thresholds reported for the v2 detection algorithm of CALIOP. The CALIOP has a very good data coverage and is providing data most of the time, but we might have some grid cells without data. In that case we assume that no PSCs have been observed. This is strictly not correct, but should not affect the overall result, since grid cells without data occur rarely.

L. 16, p.13: The formulation is confusing: is "the sum of all layers" an amount of layers or a distance in km (= amount of layers x 1.5 km) ? ANSWER: A distance in km.

Caption Figure 4: "the number of km": Please be more specific: does it concern the altitude range ? ANSWER: YES

L. 6, p.14: What do the authors mean by "NAT‐like" ? The ensemble NAT mixtures + enhanced NAT mixture ? ANSWER : YES in the text we added "NAT plus enhanced NAT"

L.1, p.17: Are there no reasons to think that it is the CALIPSO PSC frequencies that are underestimated with respect to the reality ? I have in mind the way the statistics are processed, the use of monthly means, and the characteristics of the CALIPSO/ground‐base station coverage. ANSWER: The CALIPSO observations are as close to reality as one could wish. The models are surely less "realistic".

L.4, p.17 and Figure 7: "a very large underestimation": with respect to what ? In July, it is very similar to WACCM‐cmmi, and very similar to WACCM in August. In September, LMDZrepro is much larger than WACCM. The "very large underestimation" is certainly not general when considering the total PSC frequency. However, it is true when considering the SAD criteria (Figure 7). It has to be noted that LMDZrepro gives overall the closest to CALIPSO in both cases (Total PSC frequency and SAD). Would the similarity with CALIPSO and the outlier character with respect to the other models in the case of the SAD diagnostic be related to the coarser grid resolution of the LMDZrepro model with respect to the other models ?

ANSWER: The sentence should read "The LMDZ model predicts much different NAT (June and July) and ice frequencies (all months) with respect to the other models." We have no reason to assume that the coarser grid of LMDZrepro causes the difference with other models.

L.5, p.17: "The largest biases are found for ice PSCs that tend to be significantly overestimated": Do the authors mean: "underestimated" ? I guess they are still considering the LMDZ model ? ANSWER: The sentence should read: "The largest biases are found for ice PSCs that tend to be significantly overestimated for all models except for

LMDZ, which predicts too small ice frequencies" L. 7‐8, p.17: Taking into account the difference in assumptions, what is the reliability and the robustness of such diagnostic method ? A sensivity study might be needed. ANSWER: Even if differences in the assumptions on the mean particle size may be critical, all the models have constructed and have tuned their parameterization in order to simulate a correct PSCs polar chemistry. The aim of this section is to show the variability between the CCMs in their SAD by comparing to realistic estimate of this range derived from the CALIOP observations for NAT and ICE, and not to score them. We propose this diagnostics (the range derived from observations) to be compared with the models in order to derive implications for simulated heterogeneous chemistry. Reviewer is right as a sensitivity study on instantaneous model outputs in Specified Dynamics runs would be needed to tune the proposed diagnostics and turn it into a specific set of metrics. A clarifying sentence has been added in section 3.3. L. 6, p.18: "This in turn would give less irreversible denitrification processes than in the case of simulation by the models with larger NAT SAD" ? ANSWER: What we mean here is that a smaller NAT radius would therefore give less irreversible denitrification. L.4, p.19: occurences of what ? Please be more specific. ANSWER: We mean the occurrences of the different PSC types as observed by CALIOP and simulated by the models (NAT and ice only) L. 6, p. 19: How is the averaging performed ? As a simple mean of all numbers ? Or by weighting by the grid cell area ? Concerning CALIPSO, how do the authors use the monthly means ? By making a mean of means ? Averaging yet averaged values may affect significantly the results. ANSWER: For the models the grid cells have been summed, for CALIOP the data have been gridded on a horizontal grid of 10x3.5 (lat-lon) degrees, and a vertical resolution of 1.8 km. The averages have been made by summing over all cells and months. L. 10‐12, p.19: "Too slow", "too fast": with respect to CALIPSO ? This should be specified. What do the authors mean by "progression for ice/NAT" ? ANSWER: The expressions "too fast" and "too slow" are with respect to CALIOP. The sentence "progression for ice/NAT" means that the increase of NAT and ice fractions occurs with a stronger temperature (T-TNAT) with respect to CALIOP (dashed lines in

the new figure)

L. 1, p.20: "The fraction of data with different PSC": Please revise the formulation. ANSWER: The sentence has been reformulated as follows. "The temperature dependence of the fractions of the different PSC types helps in evaluating....." L. 3, p.20: the fraction of what ? Please be specific ! "an increase of ice with TTNAT < ‐5K": Please revise the formulation: increase with decreasing temperature. ANSWER: The sentence has been reformulated as follows."The CALIOP data show a steady increase of the NAT fraction with decreasing T-TNAt up to a value of -10 K, while the increase of the ice fraction shows a higher slope belowe T-TNAT = -10 K. L.5, p.20: "a sharper increase of the fraction": fraction of what ? ANSWER: The sentence has been reformulated as follows."The increase of NAT and ice fraction for lower temperatures L. 7, p.20: "while for the other models, the ice...".

ANSWER to the previous three comments. Figure 8 has been edited to show the dependences of CALIOP also in the graphs of the models as dashed lines. This facilitates the comparison of models with CALIOP. The paragraph has been reformulated. "The onset of NAT is similar for all models, except for WACCM-ccmi, where NAT starts to form only below Tnat. The onset of the ice formation occurs at T-Tnat = -5 K for all models, except for CCSRNIES. The increase of NAT occurrences with decreasing temperatures is stronger for all models with respect to CALIOP. This is due to the fact that the models consider only the thermodynamic equilibrium conditions for the formation of PSC, and do not allow the existence of supersaturation without PSC formation. The family of models CAM3.5, WACCM and WACCM-ccmi show a faster increase of the ice occurrences with decreasing temperatures with respect to CALIOP. The reason is probably the same as for the NAT behaviour. LMDZ-repro evidently produces much less ice than the other models and CALIOP, and at low temperature NAT is the dominating species, while the other models and CALIOP show a dominant ice occurrence for low temperatures. The CCSRNIES model shows a slower increase of the ice occurrences with respect to CALIOP and the other models." 4. Conclusions. L. 12,

p.20: A point‐to‐point comparison is always feasible ! The issue is to know if it is valid and reliable. ANSWER: The referee is correct in stating that a point-to-point comparison is always feasible, but the point is if it makes much sense to do so. As has been pointed out above, many sources of biases exist and any single comparison of two observations might suffer more or less from one or more biases. So one should perform a statistical analyses on a large number of point-to-point comparison. This is not very different from our approach; we show that for short periods with many co-located observations, in particular July and August 2006. We agree with the referee that the statistics for a five year period does not confirm the agreement between the two datasets, but merely demonstrates that both instruments measure an average occurrence of all PSC types. The text has been adapted along these lines. L. 14, p. 20: "very similar": Based of the results presented in Figure 1, I don't agree. (See comment above). At least, a statistical indicator and quantitative estimates of the uncertainty should provided. ANSWER: We agree with the referee that it is preferable to consider only short periods with a good coverage of both instruments. L. 16, p.20: As already mentioned, I don't understand the emphasis on "below15 km". Is it based on Figure 2; If well, this seems very subjective to me. ANSWER:The discussion about above/below 15 km has been eliminated. It was based on a figure which proved to be wrong, due to several bugs in the normalization of the fractions ANSWER: L. 20, p.20: "Models fail to reproduce realistic geographical distributions of PSCs": I am really not convinced by the demonstration made in this paper. A significant part of the problem might come from the way the authors implement their different methodologies, and more particularly from the comparison of things that are not really comparable. ANSWER: The more symmetric distribution of PSCs in the models with respect to CALIOP is probably due to the incorrect temperatures produced by the models, since they don't include temperature fluctuations due to gravity waves. L. 22, p.20: The more recent WACCMI‐ccmi model compared better with CALIOP only for one specific diagnostic method (based on the total PSC frequency). The issues is to understand why: in view of all my previous criticisms, it might be fortuitous. ANSWER: WACCM-ccmi is really very similar

to previous versions. The better agreement s exclusively based on the temperature behaviour.

Technical corrections: L32, P2. Has been corrected L11,p3 has been corrected L2p4, acquisition has been corrected L18 P4 . done L14-20, p5, The suggestion of the referee has been followed L32, P5 done L4, P6 re-elaborated L9P6 the sentence has been eliminated because out of place L13P6 corresponding done L16P6 5-year done L18 P6 done L19 P6 induce OK L2, 6 P7 signal-to-noise substituted all over the text L8 P11, corrected L17, P11 definition TNAT CHECK !! L1.P12 this is not anymore present in the correct pdf file L6P13 ok Caption fig 4 has been corrected

Please also note the supplement to this comment:
https://www.atmos-chem-phys-discuss.net/acp-2018-589/acp-2018-589-AC2-supplement.pdf

[Figure]

[Figure]

Fig. 1.

[Figure]

**Fig. 2.**

[Figure]

Fig. 3.

[Figure]

---

## Author Comment (AC3) · 28 Nov 2018

This comment does not require a response

---

## Referee Report (RR1)

**Referee Report on "Comparison of Antarctic polar stratospheric clouds observations by ground- and satellite based lidars and relevance for Chemistry Climate Models" by Snels et al. – Second revision**

**General comments**

The clarity of the paper has significantly improved, both with the inclusion of a new figure (Figure 1) and with the revision of the text. I am also happy that the revision of Figure 2 (now, Figure 3) focussing on the single year 2006 instead of a cumulative statistics over the period 2006-2010, possibly with the correction of some bug in the algorithm, gives a much more convincing agreement between ground-based and CALIOP observations. Together with both tables giving a detailed overview of the statistics on the various PSC classes, this gives much more confidence in the statistical analysis, and in the relevance of the paper.

In the second part of the paper (comparisons between models and CALIPSO) however, the authors come back to the comparison of results covering the 5 years (2006-2010). This might be unfortunate in the sense that they loose again what they gained by restricting themselves to the year 2006 in the comparison between CALIPSO and ground-based lidar measurements above McMurdo. So, we don't know what is the effect of interannual variability on Figure 5-12, although the differences found while considering either only 2006, or 2006-2010, are significant (See Table 1), as is, consequently, the importance of interannual variability.

Further, the authors explained very well in the reply to referee that the aim of the second part of the paper is not to attribute (bad or good) scores to the different models considered here, but to propose useful diagnostic tools for the comparison between models and observations, applied here on five models. Overall , the way they revised the text reflects well this aim. However, the conclusion follows clearly the other way, electing WACCM-CCMI as "best choice". I think the authors could go beyond their current conclusion to show which kind of diagnostic they can provide on their reference model.

As a conclusion, although this paper is very interesting and, in my opinion, absolutely worth to be published as is, I think it could be improved and come to even more convincing conclusions by following the suggestions give here above.

The references to the manuscript (line and page numbers) refer to the pdf version acp-2018-589-manuscript-version5.pdf and are mentioned as indicated in this file, although the numbering suffers, from p. 12, of obvious imperfections (e.g., the first line on p. 12 is given the line number 5).

Thanks to the authors for the extended discussions provided in the reply to referee.

**Detailed comments**

**Abstract**

- L.12-13 and L. 16-20, p.1: If the aim of this work explained in the reply to the general comment ("The main focus is to define diagnostics that permit to compare observations with the "model world" in a consistent way. (…)") is now very clearly explained in Section 2.6 (L. 12-14, p. 7 and L. 10-11, p.8), the sentence in the abstract still mentions that the aim is to assess the performances of the different CCMs in simulating PSC occurrences and PSC distribution over Antarctica. It might be useful to add the clarification they provided in their response in some way in this abstract.

**2. Comparison of PSC observations by ground-based and satellite based lidars**

- The revision of Section 2.4 and the use of Figure 1 makes this section much more clear.
- L. 24-27, p.7: These two sentences are just a repetition of what is written in L. 21-23, p.7, and can thus be removed.
- L. 25-26, p.7: "in a spatial box" is not very informative; the authors could usefully repeat the dimension (I guess 7° longitude x 2° latitude). Also, "centered on" (if this is what the authors mean) could be more precise that "around".
- L. 7, p.9: 18 km might be an estimate that better reconciles the ground-based and CALIOP cases than 20 km for the lower limit of enhanced NAT mixture occurrences.
- Caption Figure 3: I guess the authors mean "the two columns".
- Caption Figure 4: "a specific temperature in arbitrary units" is surprizing. Rearranging the sentence or adding suitable punctuation might be useful to remove the confusion.
- L. 8-10, p.13: I think this sentence ("This is probably due (…) as can be seen also in figure 3") should be removed or revised. The new Figure 3 shows an ice fraction of about 20% in July with a remarkable agreement between the CALIOP and ground-based cases and cannot be used to justify the differences between both cases in Figure 4. In fact, Figure 4 shows again cumulated statistics over the years 2006-2010. It would be much better, for a better coherence and an easier and more correct comparison between the different diagnostic tools, to focus on the year 2006, also in Figure 4. This would imply, of course, that the same choice is made for the following of the paper, including the plot of Figures 5-12.

**3. Comparison of CALIOP PSC observations in the Southern Hemisphere with CCM simulations**

- L. 5 (as indicated in pdf version), p.14: "CAM3.5 and WACCM allow for saturation of up to 10 times saturation": I don't understand what the authors mean by this sentence.

- L. 10-11, p.16: "Recent studies of model simulations" is particularly vague and should be precised. Do the authors refer to "an overview of PSC simulations [by WACCM])", or to the new version WACCM/CARMA not considered here ?
- L. 8-9 (two last lines), p. 18: Do the authors take into account CALIPSO averaging kernels for this exercise ?
- L. 10-11 (as indicated in pdf version), p. 21; Figures 5-10, Table 5: There is an inconsistency between the altitude range mentioned in the figure captions on the one hand, and the Table 5 caption on the other hand.
- Caption Table 5: I don't see why the authors write that "fractions below 1% are not reported in the table". E.g. LMDZrepro estimate for September is 0.1%.
- Table 5, Table 2: Isn't it strange that the estimates provided by Table 2 for July and August 2006 are so different from the ones provided by Table 5 for July and August 2006-2010 ? In particular, neither the values nor the trend (increasing/decreasing) agree in the case of ice. Further, is there any issue in considering, also in this case only the year 2006 ? This would allow a much more detailed and interesting comparison with all results of Section 2.
- Figure 11, p. 23: Is the period considered here also 2006-2010 ? Please specify in the caption.

**4. Conclusions**

- Second paragraph: In their previous reply to referee, the authors insist on the fact that it is not the scope of the paper to give general scores to the various models, but rather to present useful diagnostic tools for the comparison between model results and a set of observations. However, scoring the 5 test-models against CALIPSO is basically what is done here, with as final conclusion that WACCM-CCMI is the "best model".  In order to follow their objective explained in the reply to referee, it might be useful, in this conclusion, to go a step forward with respect to the observation of over/underestimation of NAT frequency, anticipated or delayed onset of PSC formation etc., to try drawing some (preliminary) conclusions about the performances of the model (microphysical scheme, efficiency of the dynamics, ability to describe dynamical effects such a s mountain effects, etc.) as what is done in Section 3.4.

**Technical corrections**

- L. 11, p.2: "A variety of (…) has been proposed".
- L. 15, p.12: Duplicate "to".
- L. 27, p.12: Missing parenthesis.
- L. 8, p.13: "larger than" instead of "more larger with respect to".
- Table 4: "ice" instead of "iice".
- Caption Table 5: Is a part missing ? There is no final punctuation.

---

## Referee Report (RR2)

A 2ⁿᵈ Review of "Comparison of Antarctic polar stratospheric cloud observations by ground-based and spaceborne lidars and relevance for Chemistry Climate Models" by M. Snels et al.

<Main Comments>

The authors have considerably revised the presentation of their results since the first submission, especially on comparison of PSC statistics from CALIOP with several CCM model results. They also made several changes in the description on PSC detection and classification schemes (Section 2.3-2.5), which enables readers to understand the method used in this study well. Several typos and careless mistakes are mostly corrected as appropriate. The authors also replied to most of my major comments in the revised manuscript. I feel that the revised manuscript is almost acceptable for publication in ACP, after some minor points are revised as is pointed out below.

<Minor Comments>

(M1) P2, L8: The abbreviation of CALIOP should also be shown here, in addition to the abstract.
(M2) P2, L32: clouds and aerosol --> clouds and aerosols
(M3) P3, L10: Chemistry Climate Models --> CCMs
(M4) P15, L23-25: A reference for this sentence should be shown.
(M5) P22, L14: The reference for CCMVal-2 report, 2010 should be shown in the reference list.

---

## Author Response (AR2)

Answers to te co-editor and referees.

Co-editor: All technical corrections have been made on the revised manuscript.

Referee 1: All minor comments (M1-M5) have been addressed in the revised manuscript

Referee 2:

**Abstract.** The main focus of this work is to define diagnostics that permit to compare (CALIOP) observations with the models. It is obvious that, while comparing the different models, some show a better agreement than others for each diagnostic tool. We cannot avoid to mention this.

**Comparison of PSC observations by ground-based and satellite based lidars.**

All suggestions have been followed and the text has been changed accordingly and some phrases have been removed.

For what concerns the comparison between model simulations and CALIOP, we want to stress that we use model output of FREE RUNS. This implies that the output is not nudged, that means that we cannot compare year to year, and interannual comparison makes no sense.

**Comparison of CALIOP PSC observations in the Southern Hemisphere with CCM simulations**.

The phrase about the supersaturation was taken from Morgenstern (2010). However we did not find a clear explanation in this references, neither in the references in Morgenstern, so we decided to eliminate the sentence.

L10-11, p16. The sentence has been modified

L8-9 No

L10-11 The inconsistency has been removed

Caption table 5. The caption has been corrected

Table 5 vs Table 2. Please Note that Table 2 refers to the CALIOP observations inside the 7x2 degrees box centered on McMurdo, while Table 5 refers to the Southern Hemisphere

**Conclusions.** The primary goal is to develop diagnostics which allow to compare observations with CCM output. In the course of comparison one cannot avoid to discuss the degree of agreement between the individual models and the observations. It is obvious that the WACCM-CCMI model in most diagnostics compares well with the observations.

**Technical corrections:**

All corrections have been made in the revised manuscript

We thank the co-editor and both referees for their constructive remarks which have much improved the paper.